# Tuning of delta-protocadherin adhesion through combinatorial diversity

Adam J Bisogni[1], Shila Ghazanfar[2], Eric O Williams[1,3], Heather M Marsh[1], Jean YH Yang[2], David M Lin[1]*

[1]Department of Biomedical Sciences, Cornell University, Ithaca, United States; [2]School of Mathematics and Statistics, The University of Sydney, Sydney, Australia; [3]Department of Biology and Chemistry, Fitchburg State University, Fitchburg, United States

**Abstract** The delta-protocadherins (δ-Pcdhs) play key roles in neural development, and expression studies suggest they are expressed in combination within neurons. The extent of this combinatorial diversity, and how these combinations influence cell adhesion, is poorly understood. We show that individual mouse olfactory sensory neurons express 0–7 δ-Pcdhs. Despite this apparent combinatorial complexity, K562 cell aggregation assays revealed simple principles that mediate tuning of δ-Pcdh adhesion. Cells can vary the number of δ-Pcdhs expressed, the level of surface expression, and which δ-Pcdhs are expressed, as different members possess distinct apparent adhesive affinities. These principles contrast with those identified previously for the clustered protocadherins (cPcdhs), where the particular combination of cPcdhs expressed does not appear to be a critical factor. Despite these differences, we show δ-Pcdhs can modify cPcdh adhesion. Our studies show how intra- and interfamily interactions can greatly amplify the impact of this small subfamily on neuronal function.

DOI: https://doi.org/10.7554/eLife.41050.001

## Introduction

The delta-protocadherins (δ-Pcdhs) are a nine-member subfamily of the cadherin superfamily (*Hulpiau and van Roy, 2009*; *Nollet et al., 2000*), and play diverse roles during neural development. Mutagenesis studies have shown that individual δ-Pcdhs are important for neural development, including hindbrain formation, axon guidance, and synaptogenesis (*Cooper et al., 2015*; *Emond et al., 2009*; *Hayashi et al., 2014*; *Hoshina et al., 2013*; *Leung et al., 2013*; *Light and Jontes, 2017*; *Uemura et al., 2007*; *Williams et al., 2011*). In humans, mutations in *PCDH19* are the causative basis of one form of epilepsy (*Dibbens et al., 2008*), and other δ-Pcdhs are implicated in various neurological disorders (*Chang et al., 2018*; *Consortium on Complex Epilepsies, 2014*; *Morrow et al., 2008*).

How does this relatively small gene family mediate these varied effects? While significant effort has been devoted towards characterizing the role of individual δ-Pcdhs in neural development, almost nothing is known regarding how multiple family members function together. The δ-Pcdh subfamily has been further divided into the δ−1 (*Pcdh1*, *Pcdh7*, *Pcdh9*, and *Pcdh11x*) and δ−2 (*Pcdh8*, *Pcdh10*, *Pcdh17*, *Pcdh18*, and *Pcdh19*) subfamilies based on differences in the number of extracellular domains and also within the intracellular domain (*Redies et al., 2005*; *Vanhalst et al., 2005*). Double label RNA *in situ* hybridization studies indicate individual neurons express more than one δ-Pcdh (*Etzrodt et al., 2009*; *Krishna-K et al., 2011*). This suggests a model where different combinations of δ-Pcdhs may be expressed within different populations of neurons. Whether such combinations exist or how many δ-Pcdhs may be expressed per neuron is still not known. It seems reasonable, however, to postulate that combinatorial expression would greatly enhance the impact

*For correspondence:
dml45@cornell.edu

**Competing interests:** The authors declare that no competing interests exist.

**eLife digest** Multicellular life depends on cells being able to stick together. The human body, for example, consists of trillions of cells grouped into tissues and organs. The brain alone contains some 87 billion neurons organized into complex networks. To stay together, cells use proteins on their surface called cell adhesion molecules (CAMs). There are four major families of CAMs, each with multiple members, and the CAMs on one cell recognize and interact with the CAMs on another.

But how does this process work? One possibility is that different combinations of CAMs allow different cells to stick together. Bisogni et al. tested this idea by studying a family of CAMs called the delta-protocadherins. This family has nine members, each with its own gene. Before cells can use a gene to produce a protein, they must first use the gene's DNA as a template to build an RNA molecule. By counting the number of different types of RNA molecules inside individual cells, Bisogni et al. showed that sensory neurons in the mouse each produce up to seven different delta-protocadherins.

Further experiments revealed that cells fine-tune their interactions by varying the number, type and combination of delta-protocadherins on their surface. In addition, the delta-protocadherins also alter interactions between members of a related gene family, the clustered protocadherins. This further increases their ability to regulate how cells interact.

In contrast to previous studies that focused on single molecules, Bisogni et al. have shown how combinations of molecules work together to influence cell adhesion. Deciphering this combinatorial code is key to understanding how interactions between cells go awry in disease.

Mutations in the genes for CAMs often impair brain development. The reported findings may provide insights into how such mutations disrupt the CAM combinatorial code and alter cell to cell interactions.

DOI: https://doi.org/10.7554/eLife.41050.002

of δ-Pcdhs on cellular function. If such combinations exist, how they would influence or modify δ-Pcdh-mediated adhesion is also unknown.

The importance of examining intrafamily δ-Pcdh interactions was recently underscored by a study examining the role of δ-Pcdh adhesion in *PCDH19*-GCE (girls clustering epilepsy), a form of epilepsy limited to females. Pederick et al. demonstrated that mutations in *PCDH19*, a δ−2 family member, affected cell sorting in both *in vitro* aggregation assays and in brains of mice. Furthermore, they also showed that humans with *PCDH19*-GCE exhibit abnormal cortical folding patterns (*Pederick et al., 2018*). Importantly, they noted that PCDH19 is likely to be co-expressed with other δ-Pcdh family members, and tested how expressing PCDH10 and/or PCDH17 with PCDH19 affected sorting behavior in aggregation assays. In each case, the observed cell sorting behavior varied depending upon which δ-Pcdhs were co-expressed.

This study demonstrated the importance of defining intrafamily interactions in order to understand how loss of *Pcdh19* influences function. However, it did not define the extent of such combinations *in vivo*. It also did not establish any guiding principles for δ-Pcdh adhesion, or how different combinations influence adhesion.

Here, we uncover principles used by the δ-Pcdhs to regulate combinatorial adhesion. We first used single color and double label RNA *in situ* hybridization to show that olfactory sensory neurons (OSNs) are likely to express different combinations of δ-Pcdhs. We next employed single cell RNA analysis to establish the scope of these combinations, and find individual OSNs express between zero and seven δ-Pcdhs. We then systematically address the impact of this combinatorial diversity on intrafamily interactions by utilizing cell aggregation assays. In striking contrast to what has been seen for the clustered protocadherins (cPcdhs; *Thu et al., 2014*), we observed a range of potential adhesive behaviors. We were able to define fundamental principles that regulate these outcomes. In combination, these principles provide cells with a powerful means of fine tuning their adhesive interactions with other cells. Finally, we show that δ-Pcdhs can also modify the adhesive function of cPcdhs, which have been shown to be important for neuronal survival, dendritogenesis, synapse formation, and self-avoidance (*Lefebvre et al., 2012*; *Molumby et al., 2016*; *Wang et al., 2002*;

*Weiner et al., 2005*). These results provide an initial glimpse into interfamily interactions among protocadherin subfamilies. Our studies therefore provide a framework for determining how combinations of δ-Pcdhs mediate adhesion, and also lay the foundation for understanding how different cadherin subfamilies integrate to regulate cell-cell adhesion.

## Results

### Defining combinatorial expression of δ-Pcdhs in single neurons

We first performed single color RNA *in situ* hybridization to examine δ-Pcdh expression in the olfactory epithelium (*Figure 1—figure supplement 1A–G*). All detectable δ-Pcdhs were expressed in a punctate pattern, indicating differential expression among OSNs. Interestingly, the expression pattern for any given δ-Pcdh was not uniform throughout the epithelium. For example, *Pcdh1* is more highly expressed in the lateral epithelium, and more weakly medially (*Figure 1—figure supplement 1B,C*). In both regions, the expression was clearly punctate, but greater numbers of OSNs in the lateral epithelium expressed *Pcdh1*. In contrast, other δ-Pcdhs, such as *Pcdh9* and *Pcdh17*, show the opposite pattern, and are more strongly expressed medially with relatively low expression laterally (*Figure 1—figure supplement 1D–G*). Differences between δ−1 and δ−2 family members could not be distinguished based upon these patterns. These patterns are essentially maintained as development proceeds, although subtle changes in expression did occur. One exception was *Pcdh10*, whose expression we previously demonstrated to be dependent upon odorant-mediated activity (*Williams et al., 2011*).

The δ-Pcdhs are therefore expressed in regional patterns that overlap one another, suggesting combinatorial expression. We used double label RNA *in situ* hybridization to begin testing this hypothesis (*Figure 1A*). We systematically assayed all expressed pairs to show that 5–35% of olfactory sensory neurons (OSNs) co-express any two δ-Pcdhs (*Figure 1—figure supplement 1H*). Interestingly, the degree of co-expression varied within the family. For example, *Pcdh1* and *Pcdh7* were only co-expressed 8% of the time, while *Pcdh8* and *Pcdh9* were co-expressed 35% of the time.

OSNs expressing the same odorant receptor project to common targets within the olfactory bulb (*Ressler et al., 1994*; *Vassar et al., 1994*). Mutant analysis of members of the δ-Pcdh and cPcdh subfamilies has previously shown these genes are important for OSN targeting (*Hasegawa et al., 2008*; *Mountoufaris et al., 2017*; *Williams et al., 2011*). Interestingly, however, not all OSN populations were equally affected. Why some populations expressing a particular odorant receptor were more strongly affected in the mutant than those expressing a different receptor is unknown. We theorized that different OSN populations may express different combinations of δ-Pcdhs. Changes in these combinations would therefore affect cell adhesion mediated by the δ-Pcdhs. We therefore performed a second double label RNA *in situ* hybridization series to survey which δ-Pcdhs are co-expressed among OSNs expressing a given odorant receptor. For any one δ-Pcdh, we examined on average ~70 cells expressing a given odorant receptor to determine the degree of overlap (*Figure 1B,C*).

Confocal analysis showed all five OSN populations surveyed express varying proportions of different δ-Pcdhs (*Figure 1B,C*). There were striking differences in expression of δ-Pcdhs among the different OSN populations, arguing for the presence of specific combinations of δ-Pcdhs within each population. Interestingly, we did not find a simple one-to-one correspondence between odorant receptor expression and δ-Pcdh expression. Instead, different OSN populations varied in the proportion of δ-Pcdhs they expressed. For example, *Pcdh9* was expressed by more than half of all OSNs expressing *Olfr558*. In contrast, ~12% of *Olfr557* OSNs expressed *Pcdh9*. The variation in δ-Pcdh expression within OSN populations indicates additional levels of regulation must exist. Nevertheless, different OSN populations clearly possess differences in the proportion of δ-Pcdhs expressed by those OSNs. Such differences could be important for defining how δ-Pcdhs mediate targeting.

We next used the NanoString nCounter platform (*Geiss et al., 2008*) to more precisely define the extent of co-expression. We isolated 50 randomly selected OSNs, and performed single neuron RNA analysis for δ-Pcdhs and a subset of other genes. A heat map of the raw NanoString data showed strong heterogeneity among OSNs (*Figure 1D*, *Figure 1—source data 2*). To classify δ-Pcdhs as being 'on' or 'off' in a neuron, we used a constrained gamma-normal mixture model (*Ghazanfar et al., 2016*; *Figure 1—figure supplement 1I*). This revealed that individual OSNs

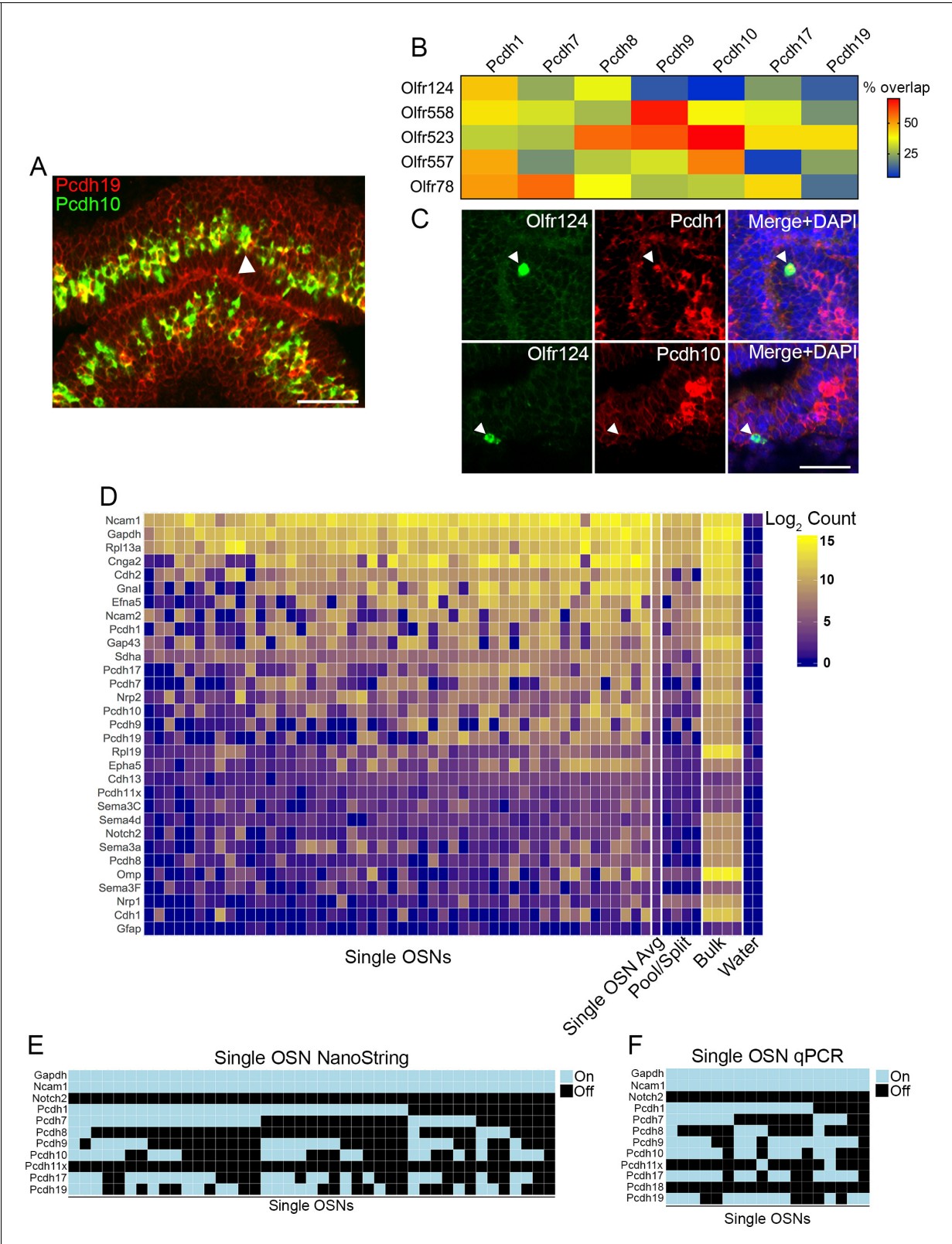

**Figure 1.** Combinatorial expression of δ-Pcdhs in mouse olfactory sensory neurons (OSNs). (**A**) Representative image of a double label RNA *in situ* hybridization with *Pcdh19* (red) and *Pcdh10* (green) in E17.5 olfactory epithelium. Both probes are co-expressed in a subset of neurons (arrowhead). Scale bar, 50 μm. (**B**) Heat map showing the percentage of co-expression among δ-Pcdhs and OSNs expressing one of five different odorant receptors. The color intensity indicates the percent of co-expression for any one δ-Pcdh with a given receptor. (**C**) Representative confocal images of *Olfr124*

*Figure 1 continued on next page*

*Figure 1 continued*

positive OSNs co-expressed with *Pcdh1* (top row) but not *Pcdh10* (bottom row). Arrowhead indicates location of *Olfr124* positive OSN. Scale bar, 50 µm. (**D**) Heat map of $\log_2$ transformed NanoString counts. (**E**) Constrained gamma-normal mixture modeling analysis shows individual, randomly selected OSNs express zero to seven δ-Pcdhs. (**F**) qRT-PCR of randomly selected single OSNs shows a mosaic pattern of δ-Pcdh expression similar to the NanoString data.

DOI: https://doi.org/10.7554/eLife.41050.003

The following source data and figure supplement are available for figure 1:

**1 —Source data 1.** NanoString codeset and primer sequences.
DOI: https://doi.org/10.7554/eLife.41050.005
**1 —Source data 2.** NanoString nCounter data.
DOI: https://doi.org/10.7554/eLife.41050.006
**Figure supplement 1.** Expression of δ-Pcdhs in OSNs.
DOI: https://doi.org/10.7554/eLife.41050.004

expressed anywhere from zero to seven δ-Pcdhs (*Figure 1E*), far exceeding prior estimates based on RNA *in situ* studies. We were unable to determine if there was any preference for co-expression among or between the δ−1 or δ−2 subfamilies.

We performed several validation experiments (see Validation of NanoString data, *Figure 1F*, and *Figure 1—figure supplement 1J*), including qRT-PCR on individual OSNs. The observed 'on' or 'off' expression pattern of this particular validation experiment was highly similar to our NanoString results (*Figure 1F*). We chose NanoString because we hypothesized a targeted approach would be more sensitive than single cell RNA-seq, which is often limited by low capture efficiency of mRNA (*Islam et al., 2011*; *Marinov et al., 2014*). Subsequent comparison with single OSN RNA-seq data sets confirmed this hypothesis (*Figure 1—figure supplement 1K,L*).

## δ-Pcdhs are homophilic cell adhesion molecules

To determine how δ-Pcdh combinations affect adhesion, we used K562 cell aggregation assays. K562 cells are commonly used to study adhesion mediated by cadherins because it is believed they do not express endogenous cadherins and are non-adherent (*Ozawa and Kemler, 1998*; *Schreiner and Weiner, 2010*; *Thu et al., 2014*)

Our initial experiments showed extracellular and transmembrane domain (ECTM) constructs were easier to express than full-length constructs. Importantly, the ECTM domain was sufficient to drive homophilic adhesion (*Figure 2—figure supplement 1A*). As our goal was to isolate the effects of adhesion on cell-cell interactions, we chose to use the ECTM domain for all subsequent experiments. As expected, the exogenously expressed protocadherins localized to sites of intracellular contact (*Figure 2—figure supplement 1B*). We also confirmed that δ-Pcdh adhesion is highly sensitive to EDTA, consistent with being members of the calcium dependent cadherin superfamily (*Figure 2—figure supplement 1C*). Although all expressed δ-Pcdhs induced cell aggregation (*Figure 2A*), *Pcdh10* formed very small aggregates relative to the others. We titrated the amount of DNA to try and normalize aggregate size (*Figure 2B*). However, varying the amount of *Pcdh10* DNA had little impact on aggregate size. We therefore excluded *Pcdh10* from further experiments.

We performed pair-wise assays by mixing cells expressing one δ-Pcdh (fused to P2A-GFP) with those expressing another (fused to P2A-RFP). We found that cells expressing the same δ-Pcdh inter-mix (*Figure 2C*, center diagonal) while cells expressing different δ-Pcdhs segregate from one another. We interpret these results to indicate that δ-Pcdh adhesion is strictly homophilic. Identical results were found for the cPcdh subfamily using the same assay (*Thu et al., 2014*).

## Mismatch coaggregation assays reveal differences in adhesion among δ-Pcdhs

To determine how combinatorial expression of δ-Pcdhs affect adhesion specificity, we next performed mismatch coaggregation assays. In these experiments, cells expressing a single δ-Pcdh are mixed with a second population of cells expressing the same δ-Pcdh plus an additional, 'mismatched' δ-Pcdh. Prior studies on cPcdhs using this approach showed that a single mismatch causes one population to segregate from the other, even when several cPcdhs are expressed in common

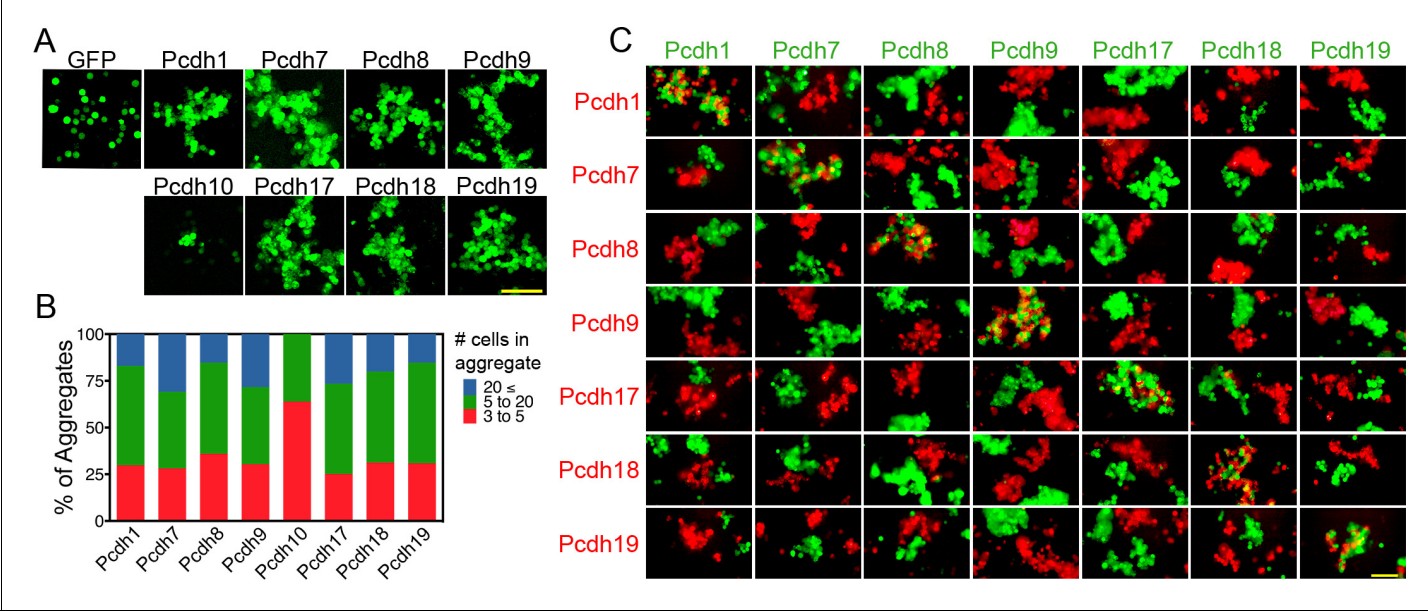

**Figure 2.** δ-Pcdhs mediate homophilic aggregation. (**A**) Aggregates formed by ECTM constructs tagged with P2A-GFP. *Pcdh11x* could not be expressed. Scale bar, 100 μm. (**B**) Distribution of aggregate sizes after titrating DNA input. Results for each δ-Pcdh were determined from three independent electroporations. *Pcdh10* aggregate size could not be increased by varying DNA input. (**C**) Pair wise analysis of δ-Pcdh binding specificity. Only pairs expressing the same δ-Pcdh coaggregated (diagonal), while cells expressing different δ-Pcdhs segregated. Results for each pair were determined from two independent electroporations. Scale bar, 100 μm.

DOI: https://doi.org/10.7554/eLife.41050.007

The following figure supplement is available for figure 2:

**Figure supplement 1.** δ-Pcdh homophilic aggregation does not require an intracellular domain and is sensitive to EDTA.

DOI: https://doi.org/10.7554/eLife.41050.008

(*Thu et al., 2014*). In contrast, this same assay suggested adhesive outcomes may be dependent on which δ-Pcdhs were co-expressed (*Pederick et al., 2018*).

To systematically define how mismatched δ-Pcdhs influence adhesive outcomes, we screened 42 possible mismatch pairs. We discovered a range of outcomes that could be grouped into three broad categories (*Figure 3A–D*). In the first, the two populations intermixed (*Figure 3A,B*). In the second, the populations interfaced (*Figure 3C*), and in the last, the populations segregated from one another (*Figure 3D*). We also noticed that interfacing and intermixing behaviors were not binary, but instead appeared to exist on a continuum.

To better capture these differences, we developed a novel metric called the CoAggregation Index (CoAg) to quantify the degree of coaggregation (see Materials and methods). Briefly, the index measures the proportion of red and green cells that share a common boundary within a given confocal image. In general, CoAg values below 0.1 indicate segregation, whereas values between 0.1–0.2 are typical of populations that interface. Above 0.2, aggregates display increasingly higher degrees of intermixing. Thus, the CoAg index captures subtle differences in aggregation behavior not easily identified by eye. Ordering the CoAg values from our screen from high to low revealed a surprisingly linear range of behavior (*Figure 3E*; mean CoAg values for a given experiment are indicated in the corner of each representative image). For comparison, the first column shows the CoAg value for *Pcdh1* cells mixed with *Pcdh7* cells (e.g. complete segregation), as expected from cPcdh mismatch assays (*Thu et al., 2014*). The red bar indicates complete mixing by matched populations.

Reordering the CoAg values into a heat map strongly argued that different δ-Pcdh combinations produced different coaggregation behaviors (*Figure 3F*). For example, we combined *Pcdh1* cells with cells expressing *Pcdh1+Pcdh7* or *Pcdh1+Pcdh8*. In the first case, cells interfaced (CoAg = 0.11; row 1, column 2), but in the second, they intermixed (CoAg = 0.27; row 1, column 3). Although *Pcdh1* was expressed by all populations, the presence of *Pcdh7* vs. *Pcdh8* led to differing behaviors.

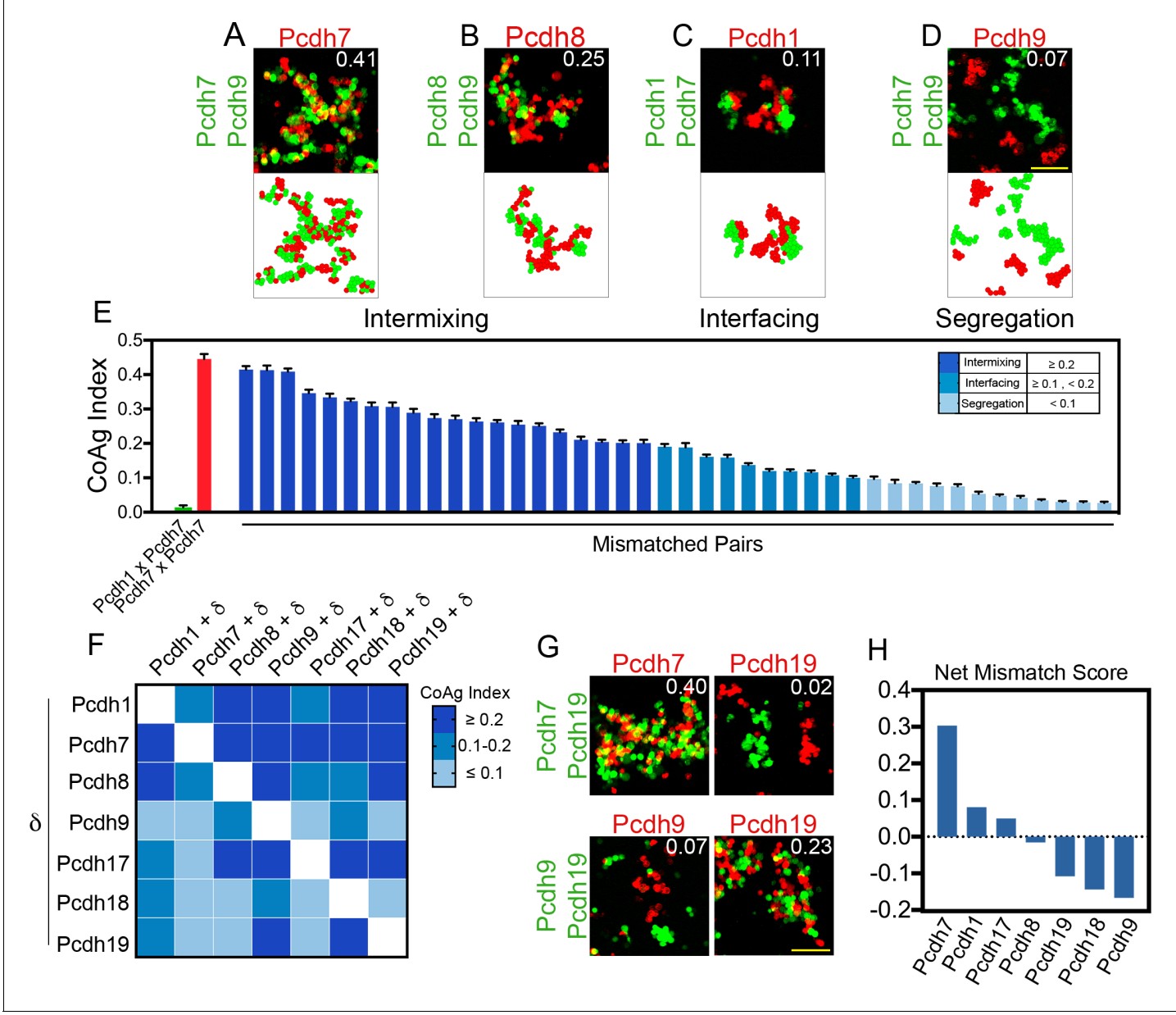

**Figure 3.** Mismatch coaggregation screen reveals complex patterns of differential adhesion. (A–D) Representative examples of different coaggregation behaviors (mean CoAg values for each experiment are displayed in the upper right of each representative image). Examples of (A) high intermixing, (B) moderate intermixing, (C) interfacing, and (D) segregation behaviors. Scale bar, 100 µm. (E) Range of coaggregation behaviors in our mismatch screen as revealed by the CoAg Index. (F) Heat map of mean CoAg values from the screen reveals high asymmetry across the diagonal. Each row represents a population expressing a single δ-Pcdh, while each column represents the cells co-expressing the listed δ-Pcdh plus the corresponding row partner. White boxes indicate redundant homophilic pairs and were not tested. Results for each of the 42 pairs tested were determined from two independent electroporations. (G) Examples of asymmetric behavior. *Pcdh7* cells intermix with *Pcdh7+Pcdh19* cells while *Pcdh19* cells segregate. *Pcdh19* cells intermix with *Pcdh9+Pcdh19* cells while *Pcdh9* cells segregate. Scale bar, 100 µm. (H) Net mismatch scores estimate the ability of a given δ-Pcdh to overcome a mismatch and still coaggregate. *Pcdh7* has the highest such score and Pcdh9 the lowest, illustrating a potential hierarchy among δ-Pcdhs.
DOI: https://doi.org/10.7554/eLife.41050.009

This suggested that, unlike the cPcdhs, the identity of the δ-Pcdh being tested is important for the outcome.

 This is further reinforced by the fact that strong asymmetry is observed across the diagonal in the heat map. For example, *Pcdh19* cells segregate from *Pcdh19+Pcdh7* cells (CoAg = 0.02; *Figure 3G*). However, 'across the diagonal,' *Pcdh7* cells intermix with these same *Pcdh19+Pcdh7*

cells (CoAg = 0.40). Similarly, *Pcdh19* cells intermix with *Pcdh19+Pcdh9* cells (CoAg = 0.23) but across the diagonal, *Pcdh9* cells segregate (CoAg = 0.07). These results strongly suggest that coaggregation is dependent upon the identity of the mismatched δ-Pcdh. We obtained similar results using full-length constructs that could be expressed to generate an aggregation behavior (data not shown). To compare how different δ-Pcdhs influence mismatch coaggregation, we generated a net mismatch score that revealed a potential hierarchy among δ-Pcdhs (*Figure 3H*, see Materials and methods).

## Differential mismatch coaggregation outcomes persist after normalizing surface expression

We next considered if these variable behaviors were caused by differential surface expression of co-expressed δ-Pcdhs. Some prior studies control for overall expression (e.g. from whole cell lysates), but not surface expression. To address this, we generated ECTM constructs fused to FLAG, GFP, or RFP, and used a cell-impermeant biotinylation reagent to label surface protein in live cells. Labeled proteins were then affinity purified and analyzed by western blotting for the various tags (*Figure 4— figure supplement 1A*). Antibody signal intensities were calibrated to allow for cross-antibody comparisons.

We re-tested all possible combinations of *Pcdh1*, *Pcdh7*, and *Pcdh17*, as these three had the strongest net mismatch scores in our initial screen (*Figure 3H*). For *Pcdh1+Pcdh7* mismatch assays, we controlled for surface expression by carefully titrating DNA input (*Figure 4A*), and examined aggregation behavior at 18, 22, 26, and 44 hr post electroporation. As seen in our initial screen, *Pcdh7* cells intermixed with *Pcdh1+Pcdh7* cells across all time points, whereas *Pcdh1* cells interfaced (*Figure 4B,C*). We used 26 hr for all further tests, given no obvious differences in behavior beyond this point.

We repeated the assay for *Pcdh1+Pcdh17*, and found that *Pcdh1* cells segregated (CoAg = 0.07, *Figure 4D–F*), while *Pcdh17* cells intermixed (CoAg = 0.42). Interestingly, these results differ from our preliminary screen, where both *Pcdh1* and *Pcdh17* cells interfaced with *Pcdh1+Pcdh17* cells. These results argue that controlling for surface level is important for interpreting coaggregation behavior, an aspect we explore below. Finally, we repeated our mismatch assay with *Pcdh7* and *Pcdh17*. We again found differences in behavior (*Figure 4—figure supplement 1B–D*). However, we found that this pair was particularly sensitive to DNA input, as small changes could alter the result despite minor effects on surface expression (*Figure 4—figure supplement 1D*). For one DNA input condition, *Pcdh17* cells interfaced (CoAg = 0.29), while in the other they segregated (CoAg = 0.08). In contrast, *Pcdh7* cells shifted towards intermixing. Nevertheless, these results confirm that differences in aggregation are dependent on δ-Pcdh identity.

Finally, we titrated surface expression for cells co-expressing *Pcdh1+Pcdh7+Pcdh17* (*Figure 4— figure supplement 1E*). We tested all 3 vs 1 (*Figure 4—figure supplement 1F*), 3 vs 2 (*Figure 4— figure supplement 1G*) and 2 vs 2 (*Figure 4—figure supplement 1H*) mismatch combinations. Differential adhesive behaviors were maintained as combinatorial depth increased, with the coaggregation outcome depending on which δ-Pcdhs were present (*Figure 4—figure supplement 1I*).

## Coaggregation behaviors can be modulated by altering relative surface expression levels

Our results argue that controlling for surface expression is important for understanding and interpreting differences in δ-Pcdh coaggregation behavior. In addition, our expression data (*Figure 1A,B* and *Figure 1—figure supplement 1A–G*) suggest that δ-Pcdh expression levels vary both within and between neurons. To further explore the role of expression, we established conditions where gradients of low, medium and high surface levels for *Pcdh1*, *Pcdh7*, and *Pcdh17* could be reproducibly generated (*Figure 5A* and *Figure 5—figure supplement 1A*). Medium levels were similar to those used in *Figure 4*.

Our mismatch assays involve mixing cells that express a single δ-Pcdh with those expressing two or more. We first asked what would happen if we altered surface expression in cells expressing a single δ-Pcdh. We found that *Pcdh1* (low, medium, and high) cells all still interfaced with *Pcdh1+Pcdh7* cells (*Figure 5B,C*), while *Pcdh7* (low, medium, and high) cells all still intermixed (*Figure 5—figure supplement 1B,C*). We found identical results with a different pair of δ-Pcdhs (*Figure 5—figure*

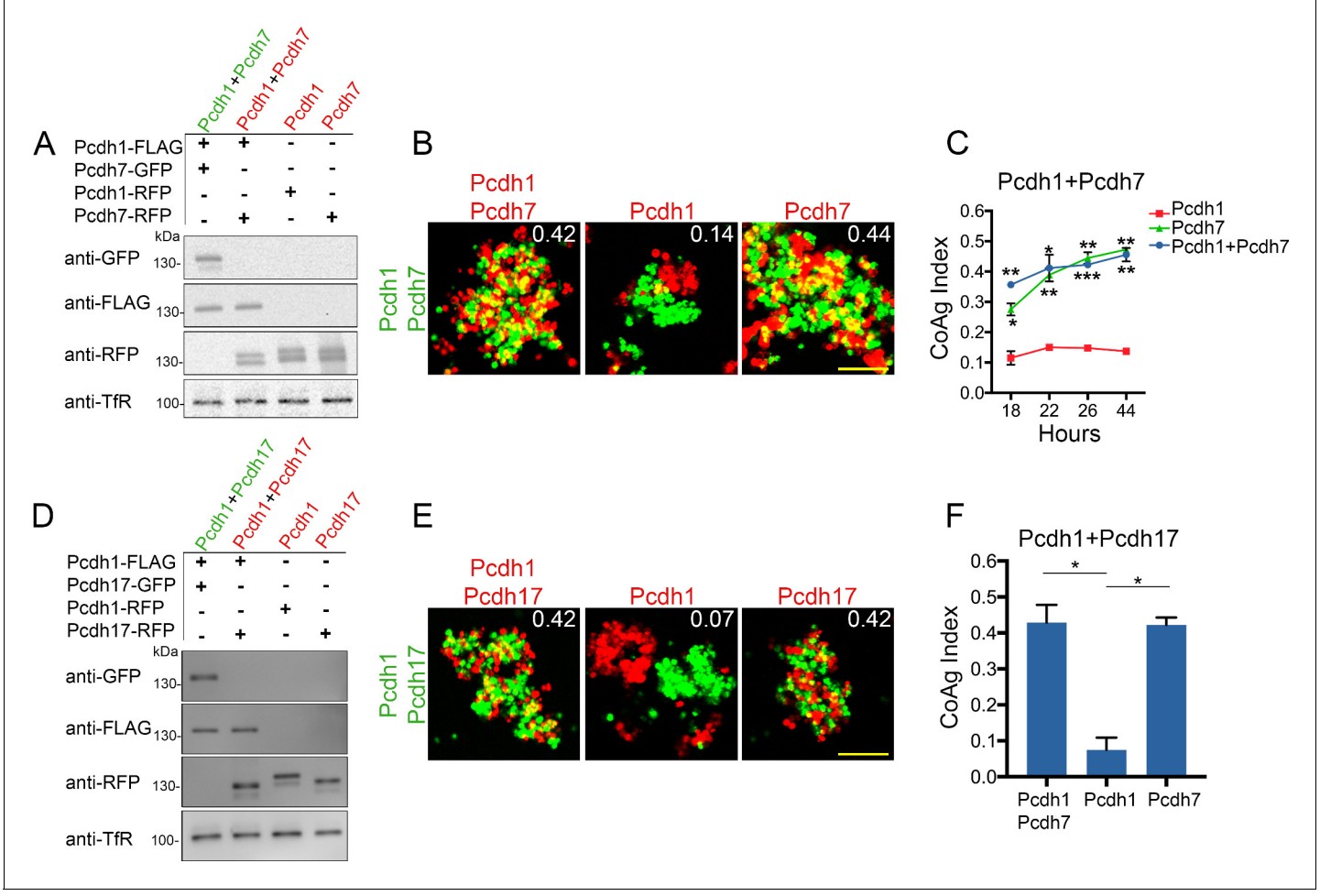

**Figure 4.** Differential coaggregation outcomes persist after controlling for surface expression levels. (**A**) Western blot of biotinylated membrane protein showing all populations in a *Pcdh1+Pcdh7* mismatch assay possess similar surface expression levels after titration. (**B**) Representative images from the mismatch assay at 26 hr. *Pcdh1* cells interface with *Pcdh1+Pcdh7* cells while *Pcdh7* cells intermix. Scale bar, 100 µm. (**C**) Mean CoAg values for each population at each time point. Each p-value is with respect to *Pcdh1*. Error bars indicate ±SEM, * indicates p≤0.05, **p≤0.01, ***p≤0.001. Results for each assay were determined from two independent electroporations. (**D**) Western blot of biotinylated membranes showing all populations in a *Pcdh1 +Pcdh17* mismatch assay possess similar levels of surface expression after titration. (**E**) Representative images from the *Pcdh1+Pcdh17* mismatch assay at 26 hr. *Pcdh1* cells segregate while *Pcdh17* cells intermix. Scale bar, 100 µm. (**F**) Mean CoAg values at 26 hr post electroporation. Error bars indicate ±SEM, * indicates p≤0.05. Results for each assay were determined from three independent electroporations.

DOI: https://doi.org/10.7554/eLife.41050.010

The following figure supplement is available for figure 4:

**Figure supplement 1.** Differences among δ-Pcdhs in coaggregation behavior remain despite controlling for surface expression levels.

DOI: https://doi.org/10.7554/eLife.41050.011

*supplement 1D–G*). While the CoAg index varied slightly, the category of coaggregation behavior (intermix, interface, or segregation) did not. Thus, differences in mismatch coaggregation among δ-Pcdhs cannot be primarily explained based on variable expression in cells expressing one δ-Pcdh.

We next asked if altering the relative proportion of δ-Pcdh expression within cells expressing two δ-Pcdhs would affect coaggregation. We created populations with high and low DNA input values for each δ-Pcdh (e.g. *Pcdh1*^High^+*Pcdh7*^Low^ and *Pcdh1*^Low^+*Pcdh7*^High^ cells). We note that our goal was to simply alter the relative proportion of surface expression in these cells, and not to establish conditions where one δ-Pcdh was necessarily higher in expression than another. We found that varying the ratio of expression clearly altered coaggregation outcomes (*Figure 5D,E*).

Differences in coaggregation behavior are most easily seen by comparing results column by column. For example, in *Figure 5D* (column 1), *Pcdh1* cells intermix with *Pcdh1*^High^+*Pcdh7*^Low^ cells, but

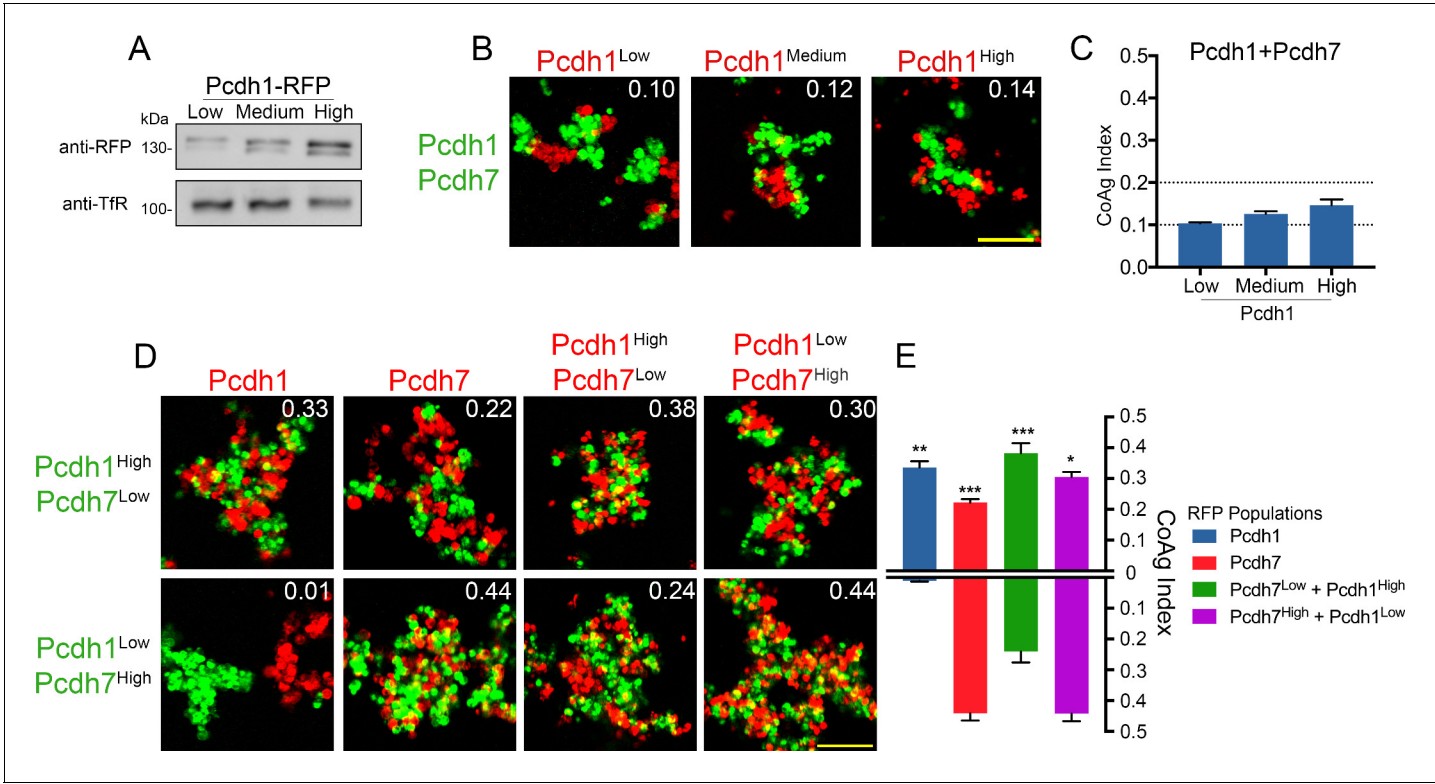

**Figure 5.** Relative surface expression modulates mismatch coaggregation behavior. (A) Western blot of biotinylated membranes showing low, medium, and high surface expressing populations of *Pcdh1* after DNA titration. (B) Representative images of mismatch coaggregation assays mixing *Pcdh1* +*Pcdh7* cells with *Pcdh1* (low, medium, and high) cells. Scale bar, 100 μm. Results for each assay were determined from three independent electroporations. (C) Mean CoAg values show varying the expression levels in *Pcdh1* cells did not alter the coaggregation behavior (interfacing), but did affect the degree of interfacing. Error bars indicate ±SEM. Dotted lines indicate thresholds for change in coaggregation category. (D) Representative images of mismatch coaggregation assays where the relative expression levels of co-expressed δ-Pcdhs were varied. *Pcdh1*High+*Pcdh7*Low cells and their complement, *Pcdh1*Low+*Pcdh7*High cells, were combined with cells expressing a given δ-Pcdh population. The two images in a given column (e.g. *Pcdh1*, column 1) illustrate the differences in coaggregation behavior when mixed with these two populations. (E) Mean CoAg values for (D), each bar indicates values for the top image in a given column vs. values for the lower image in a given column. Error bars indicate ±SEM, * indicates p≤0.05, **p≤0.01, ***p≤0.001. Results for each assay were determined from four independent electroporations.

DOI: https://doi.org/10.7554/eLife.41050.012

The following figure supplement is available for figure 5:

**Figure supplement 1.** Effects of surface expression levels on mismatch coaggregation behavior.

DOI: https://doi.org/10.7554/eLife.41050.013

segregate from *Pcdh1*Low+*Pcdh7*High cells. The coaggregation behavior of *Pcdh1* cells is therefore clearly affected by the ratio of *Pcdh1:Pcdh7* in the co-expressing cells. In the complementary experiment (column 2), *Pcdh7* cells intermixed with both *Pcdh1*High+*Pcdh7*Low and *Pcdh1*Low+- *Pcdh7*High cells. However, intermixing was clearly reduced in *Pcdh1*High+*Pcdh7*Low cells.

In column 3, *Pcdh1*High+*Pcdh7*Low cells intermixed with *Pcdh1*High+*Pcdh7*Low cells, but less well with *Pcdh1*Low+*Pcdh7*High cells. The converse (column 4) was observed for *Pcdh1*Low+*Pcdh7*High cells. Thus, relative surface levels of co-expressed δ-Pcdhs can influence aggregation behavior, even when there are no mismatches between populations.

We tested eight additional pairs using this high/low DNA input approach, and found similar results (*Figure 5—figure supplement 1H*). We confirmed a relative difference between high and low surface expression for a subset of pairs (*Figure 5—figure supplement 1I*). We conclude that changing the relative ratio of expression in cells expressing two δ-Pcdhs has a much greater effect on coaggregation than varying expression in cells expressing one δ-Pcdh.

## δ-Pcdhs possess different apparent adhesive affinities

Because differences in δ-Pcdh coaggregation behavior persisted despite controlling for surface expression, we next asked whether they possess differences in apparent adhesive affinity. Such differences have been argued to mediate segregation among classical cadherins, such as *N-* and *E-cadherin* (*Harrison et al., 2010*; *Katsamba et al., 2009*). We hypothesized that we could detect these potential differences by subjecting aggregates to higher shear forces. Cells expressing δ-Pcdhs with weaker apparent adhesive affinities should dissociate prior to those expressing δ-Pcdhs with stronger affinities.

We generated cells expressing *Pcdh1*, *Pcdh7* or *Pcdh17* at high surface levels (*Figure 5A*, *Figure 5—figure supplement 1A*), and subjected them to gradual increases in rotational speed (15–220 RPM). Images were analyzed for aggregate size using a custom written code (Aggregate Size Measurement). These populations began dissociating as speed increased. However, *Pcdh7* cells maintained larger aggregates than *Pcdh1* or *Pcdh17* cells at all speeds (*Figure 6A,B*). Furthermore, while *Pcdh1* and *Pcdh17* cells appeared to fully dissociate by ~200 RPM, *Pcdh7* aggregates were still present even at 220 RPM. Because *Pcdh1*, *Pcdh7*, and *Pcdh17* were at one end of our hierarchy (*Figure 3H*), we compared *Pcdh1* and *Pcdh19* using the same approach. Similarly, we found that *Pcdh1* cells maintained larger aggregates than *Pcdh19* cells at all speeds (*Figure 6—figure supplement 1A–C*).

Varying expression levels also accentuated these differences. We generated cells expressing *Pcdh7* or *Pcdh17* at low, medium and high levels (*Figure 5A* and *Figure 5—figure supplement 1A*). As expected, we found that higher surface levels generated larger aggregates that could better withstand increasing rotational speeds (*Figure 6—figure supplement 1D–G*). We also found that *Pcdh7* cells produced larger aggregates at all speeds compared to *Pcdh17* cells. Even at 220 RPM, *Pcdh7*$^{Low}$ cells still maintained some aggregates.

If *Pcdh1* has weaker apparent adhesive affinity than *Pcdh7*, this difference could explain why *Pcdh1* cells interface with *Pcdh1+Pcdh7* cells while *Pcdh7* cells intermix in mismatch assays. Such differences should be accentuated by increasing shear force on aggregates. To test this, we repeated the *Pcdh1+Pcdh7* mismatch assay. After allowing aggregates to form at 15 RPM, we increased the speed to 120 RPM. Despite the increased speed, *Pcdh7* cells still intermixed with *Pcdh1+Pcdh7* cells. However, *Pcdh1* cells now segregated (*Figure 6C,D*), consistent with weaker apparent adhesive affinity.

To examine structural differences that could account for this varying behavior among δ-Pcdhs, we performed multiple sequence comparison by log expectation (MUSCLE) alignments. We found low sequence identity among δ-Pcdhs in extracellular domains (EC) 1–4 (~35%; *Figure 6—figure supplement 1H*). Prior work had shown that the adhesive interface of *Pcdh19* was localized to EC1-4 (*Cooper et al., 2016*). To test the importance of EC1-4 in adhesion mediated by other subfamily members, we deleted these domains (Δ1–4) from *Pcdh1*, *Pcdh7* and *Pcdh17*. Although the truncated proteins were still transported to the surface, they were unable to mediate adhesion (*Figure 6—figure supplement 1I,J*). To determine how EC1-4 affect mismatch coaggregation, we mixed cells co-expressing *Pcdh1+Pcdh7*$^{Δ1-4}$ with those expressing *Pcdh1* or *Pcdh7* alone. *Pcdh7* cells could no longer intermix, and switched to a segregation behavior (CoAg = 0.01; *Figure 6—figure supplement 2A,B*). Conversely, *Pcdh1* cells switched from interfacing to intermixing (CoAg = 0.25). Next, we swapped the EC1-4 of *Pcdh7* with that from *Pcdh1* (*Pcdh7*$^{EC1-4:Pcdh1}$). These cells now intermixed with *Pcdh1* cells, but segregated from *Pcdh7* cells (*Figure 6—figure supplement 2C,D*, column 3). Finally, *Pcdh1* cells now intermixed with *Pcdh7*$^{EC1-4:Pcdh1}$+*Pcdh1* cells, while *Pcdh7* cells segregated (*Figure 6—figure supplement 2C,D*; column 4). These results are consistent with EC1-4 mediating adhesive specificity.

Our results argue that differences in apparent adhesive affinity and relative surface expression regulate coaggregation behavior. We therefore performed Monte Carlo simulations using a custom program (cellAggregator, *Ghazanfar, 2018*) to see if we could model these factors *in silico*. We successfully captured the behavior of a subset of our experiments. The model functioned most optimally in predicting cells that will intermix. For example, the model correctly predicted that cells expressing identical δ-Pcdhs will intermix. Furthermore, the model also predicted the behavior of cells known to intermix in mismatch coaggregation assays. However, the model could not precisely recapitulate conditions where mismatched cells interfaced or segregated (*Figure 6E*, far right column;

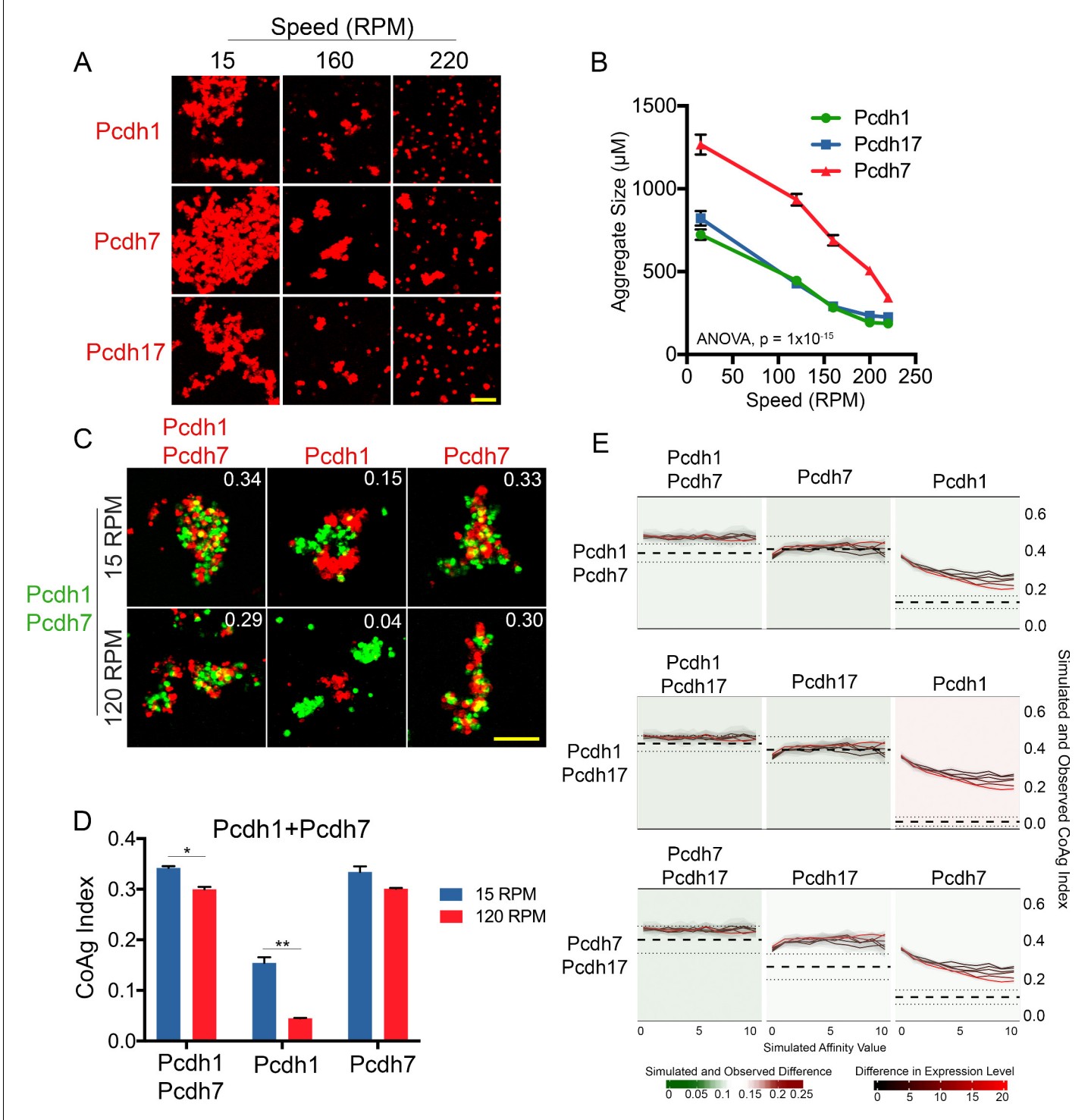

**Figure 6.** δ-Pcdhs possess differences in apparent adhesive affinity. (**A**) Representative images of cell aggregates at select speeds. *Pcdh7* cells possessed small aggregates even at 220 RPM while *Pcdh1* and *Pcdh17* cells dissociated. Scale bar, 100 μm. (**B**) Mean aggregate size at each speed. *Pcdh1* and *Pcdh17* were significantly different from *Pcdh7* by ANOVA, $p=1\times10^{-15}$. Error bars indicate ±SEM. Results for each assay were determined from four independent electroporations. (**C**) Representative images of a mismatch coaggregation assay with *Pcdh1+Pcdh7* cells. At higher speeds, *Pcdh1* cells change from interfacing to segregating (middle column), while the other two populations remain intermixed. Scale bar, 100 μm. (**D**) Mean CoAg values of (**C**). Error bars indicate ±SEM, * indicates p≤0.05, ** indicates p≤0.01. Results for each assay were determined from three independent electroporations. (**E**) Monte Carlo simulations incorporating affinity and relative expression level capture most, but not all, mismatch assay results. We modeled the behavior of a given mismatch assay (e.g. row 1, *Pcdh1+Pcdh7*). The Y-axis represents the CoAg Index (simulated (solid black and red lines)

*Figure 6 continued on next page*

*Figure 6 continued*

and observed (thick dashed line with standard error represented by thin dashed lines). Solid lines represent simulations where the relative expression level of the two δ-Pcdhs has been varied (from 1:1 to 20:1). The X-axis represents increasing differences in apparent adhesive affinity (e.g. the left most point on the X-axis represents conditions where both δ-Pcdhs are of equal apparent adhesive affinity). In all three simulated coaggregation assays, the model predicted intermixing conditions (e.g. CoAg index above 0.2), but was not able to precisely model segregation or interfacing behaviors (compare right most graph in each row against the other two).

DOI: https://doi.org/10.7554/eLife.41050.014

The following figure supplements are available for figure 6:

**Figure supplement 1.** δ-Pcdhs possess differences in apparent adhesive affinity, which appears to be mediated by EC domains 1–4.

DOI: https://doi.org/10.7554/eLife.41050.015

**Figure supplement 2.** EC1-4 mediate adhesive interactions among δ-Pcdhs.

DOI: https://doi.org/10.7554/eLife.41050.016

for example mixing *Pcdh1* cells with *Pcdh1+Pcdh7* cells). Varying affinity differences, relative expression levels, or both still did not completely capture these behaviors. We anticipate other, as yet uncharacterized effects (e.g. intracellular δ-Pcdh-δ-Pcdh interactions [*Pederick et al., 2018*]) must be incorporated into the model to better capture cell adhesive behavior.

## Increasing combinatorial δ-Pcdh expression and interactions with a cPcdh family member

Our single cell RNA analysis showed individual OSNs express up to seven δ-Pcdhs. To test the impact of increasing the number of co-expressed δ-Pcdhs on mismatch aggregation, we generated populations of cells that co-expressed *Pcdh7* with one to four additional δ-Pcdhs. To confirm changes in the relative expression of *Pcdh7* vs the other co-expressed δ-Pcdhs, we measured surface expression levels (*Figure 7A*) and performed coaggregation assays with cells expressing only *Pcdh7*. We found that each additional δ-Pcdh co-expressed with *Pcdh7* led to a corresponding decrease in the CoAg index (*Figure 7B*). *Pcdh7* only cells shifted from intermixing towards interfacing as the relative proportion of *Pcdh7* decreased. Quantification of surface expression showed that the percent of *Pcdh7* with respect to total surface expression decreased from ~50% to 25%, almost perfectly mirroring the decline in CoAg index ($R^2$ = 0.94; *Figure 7C*). We repeated the experiment with *Pcdh1*, and found a similar effect (*Figure 7—figure supplement 1A,B*). In this case, increasing the number of co-expressed δ-Pcdhs shifted the behavior of *Pcdh1* cells from interfacing to segregation.

Finally, although we have focused on how δ-Pcdh subfamily members function in combination, individual neurons are likely to co-express multiple cadherin subfamily members. How δ-Pcdhs and these other subfamily members interact is not well understood. We first confirmed that cPcdh *Pcdhb11* cells completely segregate from cells expressing δ-Pcdhs, demonstrating strict homophilic adhesion (*Figure 7—figure supplement 1C*). We then generated populations co-expressing *Pcdh7* and *Pcdhb11* at three different relative expression levels for use in mismatch coaggregation assays (*Figure 7D*). At the first two relative levels (DNA input ratio of 3:4 and 1:2), surface levels of *Pcdh7* were ~45% of total (*Figure 7E*). Under these conditions, *Pcdh7* cells strongly intermixed while *Pcdhb11* cells segregated (*Figure 7F,G*). However, at a DNA ratio of 1:4 (*Pcdh7* ~20% of total), *Pcdh7* cells still intermixed but *Pcdhb11* cells could now interface. Thus, δ-Pcdhs influence the aggregation behavior of cells expressing this particular cPcdh. This raises the intriguing possibility that the two subfamilies may work in concert to specify adhesion.

## Discussion

Our results provide a foundation for understanding how a small gene family can exert unexpectedly complex influences on cell adhesion. Despite the wide range of combinatorial expression observed within single neurons, we identified fundamental principles that help dictate intrafamily interactions. First, we found that cells can vary the number of δ-Pcdhs expressed per cell. Second, we showed that individual δ-Pcdhs possess differences in apparent adhesive affinity. Third, we further demonstrated that these differences can be modulated by varying relative surface expression levels. Together, these principles dramatically augment the range of adhesive interactions mediated by this small subfamily. Despite the fact that there are only a limited number of δ-Pcdhs, these

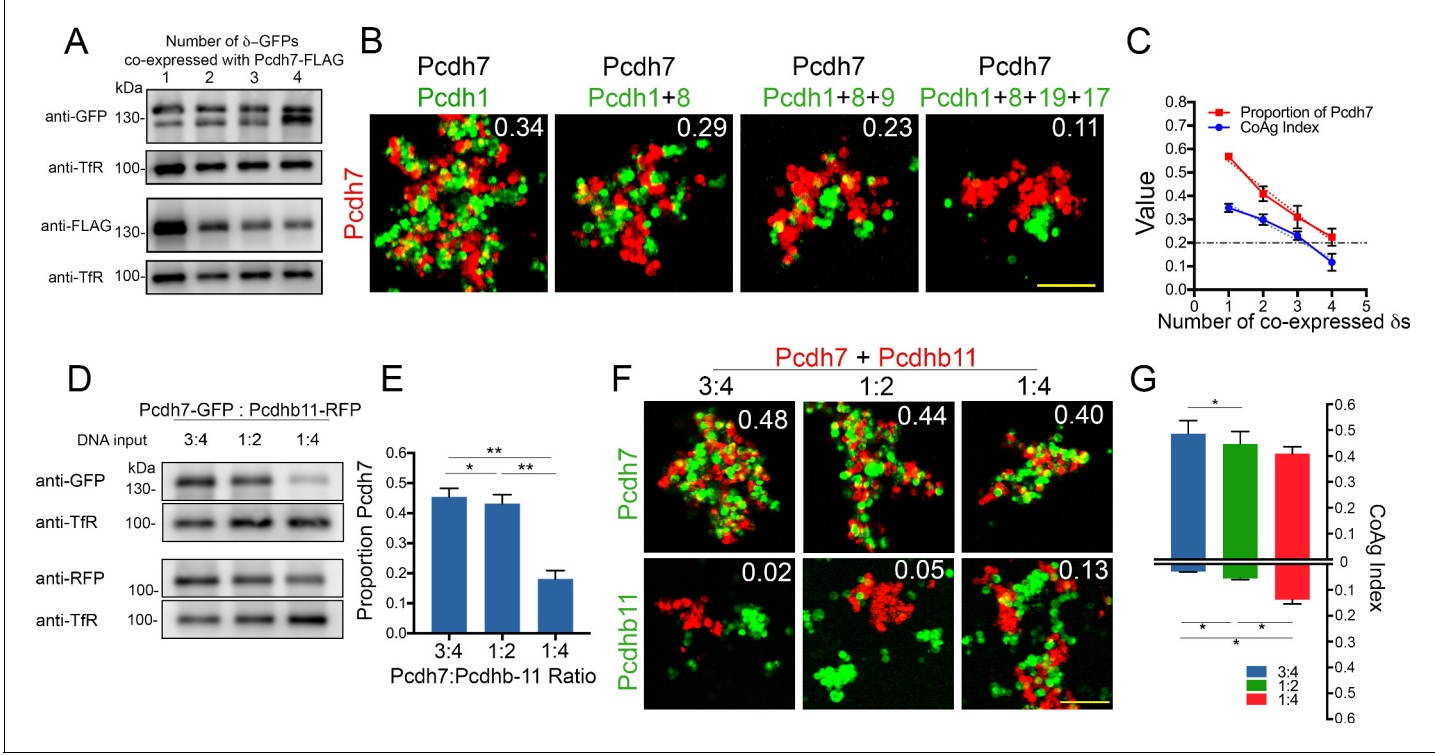

**Figure 7.** Effect of increasing co-expression of δ-Pcdhs on adhesion and interactions with clustered Pcdhb11. (**A**) Western blot showing surface expression of *Pcdh7* (FLAG) in the presence of increasing numbers of co-expressed δ-Pcdhs (all labeled with GFP). (**B**) Representative images of *Pcdh7* cells when mixed with *Pcdh7* +increasing numbers of δ-Pcdhs. Note shift from intermixing (left panel) to interfacing (right panel) as the number of δ-Pcdhs increases. Scale bar, 100 μm. (**C**) Linear regression analysis of mean CoAg values ($R^2$ = 0.94; blue) and relative surface expression of *Pcdh7* (red) with increasing numbers of co-expressed δ-Pcdhs. Error bars indicate ±SEM. Results for each assay were determined from three independent electroporations. Dot-dash line indicates boundary between intermixing and interfacing. $R^2$ = 0.97 and 0.98 for CoAg index and proportion of *Pcdh7* on surface, respectively. (**D**) Western blot of *Pcdh7* and *Pcdhb11* surface expression with varying DNA input ratios. (**E**) Quantitation of western blot data shown in (**D**). Error bars indicate ±SEM. Results for each assay were determined from three independent electroporations. (**F**) Representative images and (**G**) Mean CoAg values of coaggregation assays with *Pcdh7*+*Pcdhb11* cells. As the ratio of *Pcdh7:Pcdhb11* decreases, the CoAg value of *Pcdhb11* cells increases, and shifts from segregation to interfacing (compare bars on bottom half of graph). Although the CoAg values of *Pcdh7* drop somewhat (compare bars on top half of graph), *Pcdh7* cells still intermix, despite low DNA input ratios. Error bars indicate ±SEM, * indicates p≤0.05. Results for each assay were determined from three independent electroporations. Scale bar, 100 μm.

DOI: https://doi.org/10.7554/eLife.41050.017

The following figure supplement is available for figure 7:

**Figure supplement 1.** Increasing δ-Pcdh combinatorial expression and homophilic adhesion among protocadherins.
DOI: https://doi.org/10.7554/eLife.41050.018

principles provide cells with the ability to carefully fine tune their adhesive profiles. Even if cells express the same combination of δ-Pcdhs, varying the levels of each expressed family member provides additional flexibility in modulating adhesion. These principles contrast with those defined for the cPcdhs. However, our results also provide an initial glimpse into how these two families can interact with one another to affect adhesion.

## Differences in apparent adhesive affinity among δ-Pcdhs

The range of apparent adhesive affinities suggest that neurons can fine tune their overall adhesive profile by varying the repertoire of δ-Pcdhs expressed. One caveat is that we did not directly measure affinity using purified proteins. As our efforts are aimed at understanding how δ-Pcdhs mediate cell-cell interactions, we utilize the term apparent adhesive affinity to describe the functional impact of δ-Pcdhs on adhesion. Biophysical studies will be required to fully define such affinity differences. However, structural studies show cPcdhs possess varying adhesive affinities (*Goodman et al., 2016*;

*Rubinstein et al., 2015*). Despite this, such differences do not appear to have a major impact in K562 assays (*Thu et al., 2014*).

While cell aggregation assays have been used for decades, the technical details have never been standardized. For example, cell type, speed of rotation, time of mixing, surface expression, and mode of quantitation all differ among past studies. We note that very few studies control for or report these factors, which in our hands are important for reproducible adhesive behavior. While such controls may not be necessary when cells essentially completely segregate from one another (e.g. as for cPcdhs), such reproducibility was essential to our ability to identify and quantitate differences in adhesive outcomes among δ-Pcdh family members.

Our aggregation assay results clearly contrast with a prior study of cPcdhs (*Thu et al., 2014*). In this paper, two populations would only fully intermix if they expressed the same combinations of cPcdhs. If even one cPcdh differed between the two, the populations would completely segregate, regardless of the identity of the mismatched cPcdh. The observed results were always binary in nature, and produced either complete intermixing or complete segregation. In contrast, we were able to observe a range of coaggregation behaviors. This spectrum of adhesive outcomes illustrates how a comparatively small gene family can still have complex effects on cellular behavior. Biophysical analysis of complex formation may better illuminate the mechanism behind such differences.

We note we did not identify any obvious differences between members of the δ−1 and δ−2 subfamilies in our assays. Members of both groups were expressed in overlapping patterns within the epithelium (*Figure 1—figure supplement 1*). *In situ* hybridization, NanoString, and qRT-PCR analyses also showed no obvious differences between subfamilies (*Figure 1*). In our mismatch aggregation assays, δ−1 and δ−2 members were distributed along the spectrum of our net mismatch score (*Figure 3*). For example, *Pcdh1*, a δ−1 family member, had a roughly equivalent net mismatch score with *Pcdh17*, a δ−2 family member. However, we note that δ−1 and δ−2 members are often coexpressed within neurons, leading to potential intracellular interactions that may not be captured in these assays. Further, how the varying number of extracellular domains between the two subfamilies influence adhesion is not known. Further structural studies will be needed to better define how these differences affect cell-cell interactions.

## δ-Pcdh adhesion can be tuned by varying relative expression level

We showed a simple solution to moderating high apparent adhesive affinity δ-Pcdhs is to vary relative expression level. These results are reminiscent of principles defined for classical cadherins. Steinberg's differential adhesion hypothesis provides a commonly used framework for understanding how classical cadherins mediate cell sorting. In this model, cells sort from one another to reach an optimal thermodynamic equilibrium. This sorting can be driven by differences in adhesive affinity between cells, and/or by differences in expression level (*Foty and Steinberg, 2005*; *Friedlander et al., 1989*; *Steinberg and Takeichi, 1994*). Thus, δ-Pcdhs appear to use some of the same principles as classical cadherins. However, Steinberg and colleagues typically focused on N- and/or E-cadherin, and did not, to our knowledge, examine the behavior of multiple classical cadherins in combination. The principles we define here therefore confirm similarities between the classical and δ-Pcdhs, and extend these canonical studies of cadherin function.

We chose to use the ECTM domain for these experiments because expressing the full-length construct in K562 cells proved practically difficult. However, we demonstrated that the ECTM domain mediated homophilic adhesion to a degree similar to that of the full-length construct (*Figure 2—figure supplement 1*). As our goal was to study adhesive interactions among co-expressed family members, this allowed us to separate adhesion from intracellular signaling. In addition, the ECTM domain is typically used to study δ-Pcdh adhesion (*Chen et al., 2007*; *Cooper et al., 2016*; *Emond et al., 2011*). Still, it is clear there are many aspects of δ-Pcdh function that are not addressed by this reductionist approach. Intracellular signaling events, heterologous extracellular interactions, and regulation of δ-Pcdh gene expression can all further tune the impact of δ-Pcdhs on cell-cell interactions. Indeed, our Monte Carlo simulation indicates we can capture many, but not all, behaviors associated with combinatorial expression. Most notably, not all interface or segregation behaviors could be adequately modeled (*Figure 6E*). We expect that other, uncharacterized intracellular or extracellular interactions may explain these differences. In particular, Pederick et al. showed δ-Pcdhs can interact in *cis* (*Pederick et al., 2018*). Such *cis* interactions have previously been proposed to be critical for cPcdh function (*Rubinstein et al., 2017*; *Thu et al., 2014*). If these *cis*

interactions are also important for δ-Pcdh function, we anticipate that they may contribute towards adhesion of δ-Pcdhs in *trans*.

Nevertheless, our studies lay the foundation for new models that can integrate these principles with those defined for other cadherin subfamilies, ultimately leading to a more complete determination of cadherin function within the nervous system. Our results represent a functional genomic approach towards understanding how combinations of cadherin expression identified via transcriptomic approaches impact cellular function.

## Implications for δ-Pcdh function *in vivo*

Our reductionist approach to understanding δ-Pcdh function has the fundamental advantage of allowing us to systematically test different combinations for their impact on adhesion. Such studies would be extremely difficult to execute *in vivo*, given the varied chromosomal locations of δ-Pcdhs and the technical complexity of manipulating multiple genes at once. Further, although K562 cells have been used extensively to study protocadherin function, they are not a neuronally derived line. An appropriate question would be to ask how our results apply towards understanding δ-Pcdh function *in vivo*.

We believe there are two major applications of this study for understanding δ-Pcdh function. First, while δ-Pcdhs have been suspected to be expressed in combination *in vivo* based on double-label RNA *in situ* data, there has been no prior evidence demonstrating the extent of this expression. Our single cell NanoString and qRT-PCR data (*Figure 1D–F*) clearly demonstrate that multiple δ-Pcdhs are expressed per neuron, and show the variety and extent of such expression. Our round-robin RNA *in situ* hybridization studies (*Figure 1—figure supplement 1H*) are also consistent with this combinatorial expression. Further, our study of δ-Pcdh and odorant receptor overlap showed OSNs known to project to different targets clearly express different proportions of δ-Pcdhs (*Figure 1B*). While the expression of δ-Pcdh vs. a given odorant receptor is not a simple, one-to-one correlation, there nevertheless were clear differences among OSNs expressing different odorant receptors. Thus, the combinatorial expression of δ-Pcdhs is not an entirely random event, as has been suggested for the cPcdhs (*Goodman et al., 2016*; *Hirano et al., 2012*). This is further supported by our single label RNA *in situ* studies, which clearly shows spatially restricted expression of δ-Pcdhs within the olfactory epithelium (*Figure 1—figure supplement 1B–G*). Our results therefore demonstrate that δ-Pcdhs are combinatorially expressed *in vivo*, that 0–7 family members can be co-expressed within OSNs, and that this expression pattern is not stochastic.

Second, our studies addressed the question of how these combinations could influence δ-Pcdh function. Our results argue that the particular combination expressed within a cell has a major impact on its adhesive profile. We therefore predict mutations in any one δ-Pcdh will not have uniform effects on all cells that express that particular δ-Pcdh, simply because different cells are likely to express different combinations. For example, we previously showed that mis- and over-expression of *Pcdh10* in the olfactory system caused defects in glomerular target formation by OSNs expressing the *Olfr9* odorant receptor, but not by those expressing *Olfr17* (*Williams et al., 2011*). A recently generated *Pcdh19* mutant mouse in our lab also shows targeting defects of a subset of OSN populations (data not shown). If *Pcdh10* and *Pcdh19* are expressed by multiple OSN populations (*Figure 1B*), why are only a subset of OSNs affected in these mutants?

We speculate that this variation is due in part to the interactions between the mutated δ-Pcdh and the other, co-expressed δ-Pcdhs within a neuron. Furthermore, the two populations may express different levels of *Pcdh19*, leading to different effects when *Pcdh19* is mutated. A true understanding of how mutations in δ-Pcdhs mediate their effects would therefore be dependent on defining at a minimum what other δ-Pcdhs are co-expressed within affected cells. Loss of any one δ-Pcdh would alter the combination of δ-Pcdhs expressed and change the relative expression of co-expressed protocadherins. The changes that would occur as a result of these intrafamily interactions would therefore vary based on what δ-Pcdhs were co-expressed within the cell.

This same K562 assay was used to examine a mouse mutant of *Pcdh19* to understand why apparent cell sorting defects occurred in the cortex (*Pederick et al., 2018*). Critically, this study postulated that co-expressed δ-Pcdhs might influence the observed sorting behavior. They found that K562 cell adhesion was indeed affected by different δ-Pcdh combinations. Although they did not correct for surface expression or draw any particular conclusions about principles that mediate their observed

phenotypes, their results are consistent with ours in demonstrating the integral role of combinations in cell sorting.

Our results therefore emphasize the importance of understanding what combinations exist within neurons in order to understand observed phenotypes. However, defining the particular combination of δ-Pcdhs expressed per neuron has been problematic. Single cell RNA-seq studies have been unable to adequately address what combinations are expressed within individual neurons. Our own analysis of three single OSN RNA-seq datasets (*Hanchate et al., 2015*; *Saraiva et al., 2015*; *Tan et al., 2015*) shows an average detection of ~1 δ-Pcdh per neuron, while our NanoString approach detects ~3.5 (*Figure 1—figure supplement 1K,L*). Furthermore, our NanoString results were consistent with orthogonal validation assays using qRT-PCR and *in situ* hybridization. Thus, higher sensitivity approaches, similar to those used here, may be necessary to fully address what combinations are present within neurons.

We would also like to highlight the importance of potential, interfamily interactions. We demonstrated co-expression of *Pcdh7* with *Pcdhb11* inhibits *Pcdhb11* from intermixing with *Pcdh7 +Pcdhb11* cells (*Figure 7F,G*). If, however, expression of *Pcdh7* is reduced relative to *Pcdhb11*, then these cells begin to display interfacing behavior. Thus, δ-Pcdhs can modify the behavior of other, co-expressed subfamily members. It seems reasonable that δ-Pcdhs, classical cadherins, cPcdhs, and other subfamily members are all likely to be co-expressed within individual neurons. How would interfamily interactions influence neuronal behavior *in vivo*?

Studies on cPcdhs have emphasized the sheer number of possible stochastic combinations that can be generated with this family. Our studies demonstrate that even greater adhesive complexity can be generated by superimposing the effects of δ-Pcdhs on cells expressing cPcdhs. Although we and others have begun establishing rules governing intrafamily interactions, it is likely that further complexity can be added via interactions between subfamilies. For example, δ-Pcdhs can bind and regulate classical cadherins (*Chen and Gumbiner, 2006*; *Chen et al., 2009*; *Emond et al., 2011*). Such interfamily interactions may well help to explain certain mutant phenotypes associated with the cPcdhs. In the retina, deletion of cPcdhs leads to neuronal death and to defects in dendritic self-avoidance. Interestingly, interactions between cPcdh subfamilies accentuates these effects (*Ing-Esteves et al., 2018*), again underscoring the impact of combinatorial subfamily interactions. However, in the cortex, deletion of cPcdhs disrupts dendritic branching due to a failure to promote arborization (*Molumby et al., 2016*). Thus, the same family has distinct effects in different regions of the nervous system. These differences were proposed to be due to context dependent effects. However, it is conceivable that interfamily interactions, such as those between the δ-Pcdhs and the cPcdhs, may also play a role in explaining these varying phenotypes. The fundamental principles defined here therefore enable new hypotheses to be generated regarding how mutations in protocadherins influence neuronal function.

## Materials and methods

**Key resources table**

| Reagent type (species) or resource | Designation | Source or reference | Identifiers | Additional information |
|---|---|---|---|---|
| Gene (Mus musculus) | Pcdh1 | this paper | NCBI: NM_029357.3 | cloned from isolated RNA from mouse olfactory epithelium |
| Gene (Mus musculus) | Pcdh7 | this paper | NCBI: NM_001122758.2 | cloned from isolated RNA from mouse olfactory epithelium |
| Gene (Mus musculus) | Pcdh8 | this paper | NCBI: NM_001042726.3 | cloned from isolated RNA from mouse olfactory epithelium |
| Gene (Mus musculus) | Pcdh9 | this paper | NCBI: NM_001271798.1 | cloned from isolated RNA from mouse olfactory epithelium |

*Continued on next page*

*Continued*

| Reagent type (species) or resource | Designation | Source or reference | Identifiers | Additional information |
|---|---|---|---|---|
| Gene (Mus musculus) | Pcdh10 | this paper | NCBI: NM_001098172.1 | cloned from isolated RNA from mouse olfactory epithelium |
| Gene (Mus musculus) | Pcdh11x | this paper | NCBI: NM_001271809.1 | cloned from isolated RNA from mouse olfactory epithelium |
| Gene (Mus musculus) | Pcdh17 | this paper | NCBI: NM_001013753.2 | cloned from isolated RNA from mouse olfactory epithelium |
| Gene (Mus musculus) | Pcdh18 | this paper | NCBI: NM_130448.3 | cloned from isolated RNA from mouse olfactory epithelium |
| Gene (Mus musculus) | Pcdh19 | this paper | NCBI: NM_001105246.1 | cloned from isolated RNA from mouse olfactory epithelium |
| Gene (Mus musculus) | Pcdhb11 | this paper | NCBI: NM_053136.3 | cloned from isolated RNA from mouse olfactory epithelium |
| Strain, strain background (Mus musculus) | FVB/NJ | The Jackson Laboratory | 1800 | |
| Strain, strain background (Mus musculus) | C57BL/6J | The Jackson Laboratory | 664 | |
| Strain, strain background (Mus musculus) | CD-1 | Charles River | 22 | |
| Cell line (Homo sapiens) | K-562 | ATCC | CCL-243 | |
| Biological sample (Mus musculus) | primary olfactory sensory neurons | this paper | | isolated for single cell analysis from P6-P8 mice, both sexes |
| Biological sample (Mus musculus) | olfactory epithelium | this paper | | isolated and sectioned for RNA *in situ* hybridization, at ages indicated in paper, both sexes |
| Antibody | moust anti-GFP | Thermo Scientific | MA5-15256 | 1:4000 |
| Antibody | mouse anti-RFP | Thermo Scientific | MA5-15257 | 1:2000 |
| Antibody | mouse anti-FLAG | Thermo Scientific | MA1-91878 | 1:6000 |
| Antibody | mouse anti-Transferrin Receptor (TfR) | Thermo Scientific | 13–6800 | 1:1000 |
| Recombinant DNA reagent | N1-p2a-GFP or RFP | this paper | | modified from Clontech N1-eGFP |
| Recombinant DNA reagent | N1-GFP or RFP | this paper | | modified from Clontech N1-eGFP |
| Sequence-based reagent | | this paper | | see supplemental tables for all primers |
| Commercial assay or kit | Pierce Cell Surface Isolation Kit | Thermo Scientific | 89881 | |
| Commercial assay or kit | Ingenio Electroporation Kit | Mirus | MIR 50118 | |

*Continued*

| Reagent type (species) or resource | Designation | Source or reference | Identifiers | Additional information |
|---|---|---|---|---|
| Chemical compound, drug | Valproic acid sodium salt | Sigma-Aldrich | P4543 | 4 µM |
| Software, algorithm | Co-Ag index | this paper | | code written in Mathematica (Wolfram Research) |
| Software, algorithm | Aggregate size measurement | this paper | | code written in Mathematica (Wolfram Research) |
| Software, algorithm | Cell aggregation Monte Carlo Simulator | this paper | | https://github.com/shazanfar/cellAggregator ; *Ghazanfar et al., 2016* |

## Animal use

All animal protocols were approved by the Cornell Institutional Animal Care and Use Committee. Non-Swiss Albino (NSA) mice of mixed sex were used for all single cell studies. For RNA *in situ* hybridization experiments, both NSA and C57Bl/6 mice were used. Mice were sacrificed at post-natal day 7 (P7) for single cell and single label RNA *in situ* hybridization experiments, and embryonic day 17.5 (E17.5) for double label experiments.

## RNA *in situ* hybridization and quantification

Single and double label RNA *in situ* hybridization was performed essentially as described (*Williams et al., 2011*). For single color studies at E17.5 and P7, at least three independent heads were analyzed. For δ-Pcdh co-expression studies, three replicates were performed from three different heads for each gene. Imaging of double-label RNA *in situ* data was performed using a Zeiss (Wetzlar, Germany) LSM 510 confocal microscope, and multiple locations within each E17.5 olfactory epithelia were examined. Five optical slices (each 3 µm thick) from each location were used to assess co-expression. Positive co-expression was manually determined based on overlapping fluorescence signal observed in consecutive optical sections. Between 71 and 167 cells were analyzed per double label comparison. To quantify single label RNA *in situ* data, slides were scanned with a ScanScope (Leica) using a 20x objective. The OSN layer of each section was manually traced using HALO software (Indica Labs, Corrales, New Mexico), and the percent positive area was determined using a built-in software module. For δ-Pcdh and odorant receptor co-expression studies, an average of 70 OSNs expressing a given odorant receptor were analyzed for co-expression with any one δ-Pcdh.

## Single OSN isolation

Olfactory epithelia were dissected from P7 NSA mice and enzymatically dissociated for 1 hour using the Papain Dissociation Kit (Worthington, Lakewood, NJ). The tissue was manually triturated, and the papain neutralized as per manufacturer's instructions. Approximately 250,000 cells were then plated on coverslips coated with poly-ornithine, and the cells were allowed to recover at 37°C with 6% $CO_2$ for 30 min in Modified Eagle's Medium (MEM). After recovery, the cells were gently washed three times with $CO_2$ equilibrated MEM. The coverslip was then transferred to a 10 cm dish, where it was immobilized by applying small dabs of autoclaved Vaseline between the bottom of the coverslip and the 10 cm dish. The dish was flooded with 10 mL of equilibrated MEM, and individual OSNs isolated by manual aspiration under a 20X objective using a micromanipulator (Eppendorf; Hauppauge, New York). Micropipettes for aspiration were prepared using a Sutter P-97 Flaming/Brown (Novato, CA) micropipette puller, and pre-filled with ~3 µL of MEM. After aspiration, the contents were transferred to a PCR tube by gently snapping the distal tip of the micropipette inside the tube and expelling the contents using a needle and syringe. Two different lysis buffers were utilized (Cells-to-Ct or CellsDirect, Thermo-Fisher, Waltham, MA), with no apparent difference in lysis quality or NanoString results. Each tube was pre-loaded with 6 µL of CellsDirect lysis buffer (containing lysis

enhancer) or Cells-to-Ct buffer (containing DNAse I). As OSN isolation was performed at room temperature, neurons were collected from a given coverslip within 30 min. Cells processed in CellsDirect buffer were stored at −80°C until processing. Cells processed in Cells-to-Ct buffer were vortexed and then incubated at room temperature for five minutes. An additional 0.5 μL of stop solution was added and incubated for 2 min at room temperature before being stored at −80°C until further processing.

## Amplification and quality control of single OSNs

Amplification reactions were done using the CellsDirect kit (Thermo-Fisher) essentially according to manufacturer's instructions, with the following modifications. The 31 gene multiplex primer set was added to individual lysates (100 nM final) in a final volume of 10 μL. Tubes were heated at 80°C for 10 min and chilled on ice for 3 min. 10 μL of 2x reaction buffer and 1 μL of SuperScript III/Platinum Taq (Thermo-Fisher) were added and tubes were reacted in a PCR machine at 50°C for one hour, followed by 85°C for 15 min to inactivate the reverse transcriptase. PCR amplification was then performed with an initial activation at 94°C for 2 min, followed by 18 cycles of 94°C for 30 s, 60°C for 30 s, and 72°C for 30 s. After amplification, 20 μL of 10 mM Tris 7.5 was added to each sample to bring up the total volume to 40 μL. Four μL of each sample was then screened by quantitative PCR to determine expression levels of *Gapdh* (indicating successful capture and amplification) and *Ncam1* (indicating an OSN). Taqman primers were designed to amplify regions internal to the 31 gene multiplex primer sequence, and samples were run on an ABI 7500 (Thermo-Fisher). Only cells with Ct values $\leq 25$ for both genes were used for the NanoString analysis (Seattle, WA). See *Figure 1— source data 1*.

## NanoString nCounter processing and validation

A custom codeset of 31 genes was designed that would detect a select subset of known axon guidance genes (see *Figure 1—source data 1*). Single cell cDNA was hybridized to the codeset in collaboration with NanoString. Genes were determined to be positively expressed using a constrained gamma-normal mixture model approach (*Ghazanfar et al., 2016*). Briefly, 'negative' control genes (e.g. *Notch2*, *Gfap* and *Cdh13*) were used to estimate the distribution of the no or lowly expressed genes across all cells. Following this, for each cell a constrained gamma-normal mixture model was fit using the Expectation Maximization (EM) algorithm, constrained in the sense that the mean and variance of the no or lowly expressed component for that particular cell was the same as across all cells, allowing the highly expressed component to vary as required. This constrained gamma-normal mixture model allowed for 'sharing' of information across multiple cells, reducing the possibility of ill-fitting distributions to the cells' expression patterns. Following model fitting, cells and genes were classed as 'expressed' if the corresponding posterior probability was 0.5 or above, and 'not expressed' otherwise. After this analysis, some cells were found to be *Notch2* positive, and discarded from further study. Data from four codeset genes generated no useful information and were not utilized.

## Single cell qPCR validation

OSNs were isolated and amplified in a manner identical to those used for NanoString analysis. Two uL of amplified cDNA from each single cell were used as template for each Taqman assay (*Gapdh*, *Ncam1*, *Notch2*, and the δ-Pcdhs; *Figure 1—source data 1*). All primer sets displayed efficiencies between 93–100%, except for *Pcdh1* which had 83% efficiency (improvement was not observed with multiple primer designs). Probes were designed to bind to regions distinct from those detected with the NanoString codeset. Genes were considered 'on' if we observed a $C_t$ value less than or equal to 30.

## Plasmid construction

EGFP-N1 (Clontech) vectors were modified to incorporate the TagRFP fluorophore and/or a P2A sequence. FLAG constructs were created in a pHAN vector modified to include a FLAG sequence at the 3' terminus of the polylinker. ECTM domains of δ-Pcdhs and Pcdhb11 were then cloned into the appropriate vector.

## K562 aggregation assay

K562 cells were purchased from ATCC (ATCC CCL-243) and tested mycoplasma negative. Low passage number cells (4–10 passages) were maintained in RPMI +L glutamine with 10% calf bovine serum (Gemini Bio, Sacramento, CA). Cells were grown to a density between 250–500,000 cells/mL prior to electroporation. For the electroporation, one million cells were removed, concentrated by centrifugation, and resuspended in 100 µL of Ingenio Electroporation Solution (Mirus Bio, Madison, WI). Five to eight µg of cesium chloride or midi prepped (Omega) DNA for each δ-Pcdh to be expressed was added, and the cells electroporated using an Amaxa Nucleofector II (Lonza; program T-016, Cologne, Germany). Cells were allowed to recover for one hour at 37°C by immediate addition of 2 mLs of $CO_2$ equilibrated media. After recovering, valproic acid (VPA, 4 mM final; Sigma, St. Louis, MO) was added to promote expression. Preliminary control experiments showed VPA did not affect cell adhesion, as cells electroporated with vector only remained non-adherent up to four days. For coaggregation experiments, equal volumes of cells from a given electroporation were mixed immediately following the recovery period and placed in individual wells of a 6-well (2 mLs/well) or 24-well (0.5 mLs/well) plate. Cells were gently and continuously agitated at 15 RPM overnight in a tissue culture incubator at 37°C with 6% $CO_2$.

## Cell aggregation imaging

For initial aggregate size titration, 15–20 images were taken of each replicate using an inverted fluorescent microscope (Nikon, Tokyo, Japan) with a 10x objective. For speed and aggregate size experiments, ~6 field of views were captured at each speed for each replicate using a confocal microscope (Zeiss LSM 510) with a 5x objective. For all other aggregation experiments, ~10–15 confocal images were captured of each replicate using a 10x objective.

## CoAggregation index (CoAg)

To generate the Coaggregation Index, confocal images were analyzed using custom code ('CoAg') written in Mathematica (Wolfram Research, Champaign, IL). Briefly, each confocal image of an aggregate is parsed into squares just slightly larger than the area of a single cell. After removing all black squares from the image (those containing no cells), the remaining squares are analyzed to calculate the percent of squares that contain more than one color. As a result, cells that completely segregate from one another will have a very low CoAg index because few squares will contain more than one color. In contrast, cells that interface will have higher CoAg indices as green and red cells abut one another, while those that intermix will have the highest index.

## Aggregate size titration assay

K562 cells were electroporated and following a one hour recovery period, allowed to form aggregates at 15 RPM overnight. At 24–26 hr, images were captured of each replicate. To determine size of aggregates, images were analyzed using the particle size plugin in ImageJ. Aggregates smaller than three cells were removed from the analysis to prevent dividing cells and single cells not participating in aggregation from skewing the results. Aggregate pixel size was compared to the pixel area of one cell to approximate the number of cells per aggregate.

## Speed aggregation assay

K562 cells were electroporated and following the 1 hr recovery period, allowed to form aggregates at 15 RPM overnight. At 24–26 hr, images were captured to establish a 15 RPM baseline. Plates were then returned to the incubator and the speed increased for 1 hr to 120 RPM. Images were then acquired, and this process repeated at 160, 200 and 220 RPM. Each image was then analyzed using a custom written code ('Aggregate Size Measurement') in Mathematica (Wolfram Research, Champaign, IL) to measure the pixel size of each aggregate, and aggregate pixel size was then converted to microns.

## Statistical analyses

For mismatch coaggregation assays, paired t-tests were performed between each paired population to determine statistical significance in Prism (Graph Pad, La Jolla, CA). For aggregate speed and size analyses, analysis of variance (ANOVA) were performed in R.

## Biotinylation assay

Surface biotinylation of live K562 cells was performed using the Pierce Cell Surface Isolation Kit (Thermo-Fisher) essentially as recommended. Volume of cell resuspension was reduced to 1 mL, and an additional 150 uL of lysis buffer was added to ensure complete mixing during incubation.

## Western blot analysis

Western blots were performed by loading 8 uL (roughly 15% of the total elution from each biotinylation experiment) onto 10% SDS polyacrylamide gels. All primary antibodies used were monoclonal in origin, and carefully titrated to establish working dilutions of equivalent detection so that samples across antibodies could be compared. To achieve this, we calibrated working monoclonal concentrations with purified RFP and GFP proteins. We also electroporated cells with the same δ-Pcdh fused to different tags to optimize antibody dilution to account for variation in signal intensity. The antibodies used were mouse anti-GFP (1:4,000, Thermo-Fisher MA5-15256), mouse anti-RFP (1:2,000, Thermo-Fisher MA5-15257) and mouse anti-FLAG (1:6,000, Thermo-Fisher MA1-91878). We used the transferrin receptor (TfR) as a loading control for surface protein (1:1,000, Thermo-Fisher 13–6800). All antibodies were diluted in 20% glycerol upon receipt to promote cryostability. Estimation of band intensity was carried out using ImageJ.

## Monte carlo simulation (cellAggregator)

To investigate the aggregation behavior of cell populations expressing δ-Pcdhs of varying apparent adhesive affinities and expression, we performed Monte Carlo based simulations to describe cell binding interactions as a dynamic cell-cell network across discrete time steps using custom code (cellAggregator). Two cell populations, green (n = 25) and red (n = 25), were assigned properties of two hypothetical genes named A and B, corresponding to the coaggregation assay experiments conducted. For example, green cells could be designated as expressing high levels of A and low levels of B, and red cells as expressing low levels of B and high levels of A. The genes A and B were each also assigned binding affinities, for example, A possesses two times greater apparent adhesive affinity than B. The initial cell-cell network consists of the green and red cells as nodes in the network, and edges represent cell-cell binding interactions occurring.

For each simulation, 100 time steps were performed. At each discrete time point, the cells are mixed and allowed to bind to other cells according to a 'speed dating' set up, where the majority of cell pairs (arbitrarily set at 75%) result in a cell-cell interaction. Allowing the majority (as opposed to all cell pairs) to bind avoids oscillatory network behavior. The probability that two cells would 'speed date' increased as the Euclidean distance between the force-directed network projection onto two dimensions decreased, that is nodes more closely connected were more likely to 'speed date'. Once 'speed-dating' begins, the cell pair would bind via the genes expressed by each cell, with unbound genes selected at random with a probability corresponding to the expression level. The duration of interaction (number of time steps) depended on the identity of genes. A-B interactions persisted for only a single time step, while B-B interactions persisted for three time steps, and A-A interactions persisted for three multiplied by the affinity ratio time steps. This differential length of time for cell-cell interactions is based on the idea that non-homophilic protocadherin interactions are unstable and do not persist (A-B), and that some protocadherins may have different levels of apparent adhesive affinity, leading to more persistent or stable binding time (e.g. A-A lasts more time steps than B-B if A is assigned greater affinity than B). The green or red color of the cells did not affect the binding of cell pairs.

Instantaneous network coaggregation was measured by calculating the average proportion of different-color to same-color binding partners across all cells in the network for any one time step. Cells with no network partners were not included in this calculation. The *in silico* coaggregation behavior for the entire simulation was then determined as an average of all instantaneous network coaggregations in the simulation. This value did not include initial time steps (arbitrarily set at 25% of the 100 total time steps) to allow for the network to stabilize following the initial state of all cells being unconnected. This resulted in a single overall *in silico* coaggregation index value determined for the simulation scenario. A total of 100 time steps were simulated for each scenario, and each scenario was repeated five times. To model varying affinity between genes, the affinity values were allowed to range between 1 (same affinity) and 10.

The source code for performing the Monte Carlo simulation is available at https://github.com/shazanfar/cellAggregator (copy archived at https://github.com/elifesciences-publications/cellAggregator) and an interactive R Shiny application available at http://shiny.maths.usyd.edu.au/cellAggregator/.

## Validation of NanoString data

*Pcdh18* data was discarded due to an error in the codeset. However, *Pcdh18* was not detected by RNA *in situ* hybridization experiments in the epithelium nor in subsequent single OSN qPCR experiments. Negative controls (e.g. water or media only) showed no signal following amplification, indicating a lack of contamination. To validate the NanoString data, we first performed a 'pool-split' experiment to determine technical reproducibility. RNA from 12 single cells were pooled and then split into multiple aliquots. Each aliquot was separately amplified and processed to assess technical reproducibility. Samples showed good correlation ($R^2$ = 0.62; data not shown). Second, we asked if averaging the expression patterns from single neurons approximated the pattern seen using bulk epithelial RNA. We found strong correlation ($R^2$ = 0.65) despite the fact we only analyzed 50 cells, and bulk RNA contains neurons, glia, and other cell-types (data not shown). Finally, multiple discriminant analysis (MDA) showed that pool-split samples clustered with single cells while the water and bulk samples formed separate, discrete clusters (data not shown).

To address the concern that dissociation of whole epithelia would affect δ-Pcdh expression, we generated a proxy for *in vivo* expression by performing single color RNA *in situ* hybridization studies (*Figure 1—figure supplement 1A–G*; no signal was detected for *Pcdh11x* or *Pcdh18*). Interestingly, the pattern of expression was clearly variable among neurons, and unevenly distributed within the epithelium (*Figure 1—figure supplement 1B–G*). We used this RNA *in situ* data to estimate the proportion of OSNs that express each δ-Pcdh (*Figure 1—figure supplement 1J*; see Materials and methods). We found that our single neuron data and these *in vivo* estimates followed similar trends ($R^2$ = 0.58), suggesting dissociation did not have an appreciable impact on our NanoString data.

## Acknowledgements

We thank Mark Roberson, Holger Sondermann, and John O'Donnell for helpful discussions and advice.

## Additional information

### Funding

| Funder | Grant reference number | Author |
|---|---|---|
| National Institutes of Health | R21DC015107 | Adam J Bisogni |
| College of Veterinary Medicine, Cornell University | Seed Grant | Adam J Bisogni |
| National Institutes of Health | T32HD057854 | Adam J Bisogni |
| College of Veterinary Medicine, Cornell University | Center for Vertebrate Genomics - Scholarship | Adam J Bisogni |
| National Institutes of Health | R01DC007489 | Adam J Bisogni Eric O Williams Heather M Marsh |
| Commonwealth Scientific and Industrial Research Organisation | OCE Top Up Scholarship | Shila Ghazanfar |
| Australian Postgraduate Award | Postgraduate Award | Shila Ghazanfar |
| Australian Research Council | FT0991918 | Jean YH Yang |
| National Health and Medical Research Council | APP1111338 | Jean YH Yang |

The funders had no role in study design, data collection and interpretation, or the decision to submit the work for publication.

## Author contributions

Adam J Bisogni, Conceptualization, Data curation, Software, Formal analysis, Validation, Investigation, Visualization, Methodology, Writing—original draft, Writing—review and editing; Shila Ghazanfar, Data curation, Software, Formal analysis, Investigation, Visualization, Methodology; Eric O Williams, Conceptualization, Data curation, Supervision, Validation, Investigation, Methodology; Heather M Marsh, Investigation, Methodology; Jean YH Yang, Conceptualization, Supervision, Funding acquisition, Project administration; David M Lin, Conceptualization, Resources, Data curation, Supervision, Funding acquisition, Investigation, Visualization, Methodology, Writing—original draft, Project administration, Writing—review and editing

## Author ORCIDs

Adam J Bisogni (iD) http://orcid.org/0000-0001-5383-7048
David M Lin (iD) http://orcid.org/0000-0002-9236-8405

## Ethics

Animal experimentation: This study was performed in strict accordance with the recommendations in the Guide for the Care and Use of Laboratory Animals of the National Institutes of Health. All of the animals were handled according to approved institutional animal care and use committee (IACUC) protocols (#01-0075) of Cornell University.

## Decision letter and Author response

Decision letter https://doi.org/10.7554/eLife.41050.037
Author response https://doi.org/10.7554/eLife.41050.038

# Additional files

## Supplementary files

• Transparent reporting form
DOI: https://doi.org/10.7554/eLife.41050.019

## Data availability

All data generated or analyzed during this study are included in the manuscript and supporting files. A link to the software code is also provided.

The following previously published datasets were used:

| Author(s) | Year | Dataset title | Dataset URL | Database and Identifier |
|---|---|---|---|---|
| Tan L, Li Q, Xie XS | 2015 | Olfactory sensory neurons transiently express multiple olfactory receptors during development | http://www.ncbi.nlm.nih.gov/sra/SRP065920 | NCBI Sequence Read Archive, SRP065920 |
| Saraiva LR, Ibarra-Soria X, Khan M, Omura M, Scialdone A, Mombaerts P, Marconi JC, Logan DW | 2015 | Hierarchical deconstruction of mouse olfactory sensory neurons: from whole mucosa to single-cell RNA-seq | https://www.ebi.ac.uk/ena/data/view/ERS715983 | European Nucleotide Archive, ERS715983 |
| Hanchate NK, Kondoh K, Lu Z, Kuang D, Ye X1, Qiu X, Pachter L, Trapnell C, Buck LB | 2015 | Single-cell transcriptomics reveals receptor transformations during olfactory neurogenesis | https://www.ncbi.nlm.nih.gov/geo/query/acc.cgi?acc=GSE75413 | NCBI Gene Expression Omnibus, GSE75413 |
| Saraiva LR, Ibarra-Soria X, Khan M, | 2015 | Hierarchical deconstruction of mouse olfactory sensory neurons: | https://www.ebi.ac.uk/ena/data/view/ | European Nucleotide Archive, |

| Omura M, Scial-done A, Mom-baerts P, Marconi JC, Logan DW | | from whole mucosa to single-cell RNA-seq | ERS715985 | ERS715985 |
|---|---|---|---|---|
| Saraiva LR, Ibarra-Soria X, Khan M, Omura M, Scial-done A, Mom-baerts P, Marconi JC, Logan DW | 2015 | Hierarchical deconstruction of mouse olfactory sensory neurons: from whole mucosa to single-cell RNA-seq | https://www.ebi.ac.uk/ena/data/view/ERS715988 | European Nucleotide Archive, ERS715988 |
| Saraiva LR, Ibarra-Soria X, Khan M, Omura M, Scial-done A, Mom-baerts P, Marconi JC, Logan DW | 2015 | Hierarchical deconstruction of mouse olfactory sensory neurons: from whole mucosa to single-cell RNA-seq | https://www.ebi.ac.uk/ena/data/view/ERS715986 | European Nucleotide Archive, ERS715986 |
| Saraiva LR, Ibarra-Soria X, Khan M, Omura M, Scial-done A, Mom-baerts P, Marconi JC, Logan DW | 2015 | Hierarchical deconstruction of mouse olfactory sensory neurons: from whole mucosa to single-cell RNA-seq | https://www.ebi.ac.uk/ena/data/view/ERS715987 | European Nucleotide Archive, ERS715987 |
| Saraiva LR, Ibarra-Soria X, Khan M, Omura M, Scial-done A, Mom-baerts P, Marconi JC, Logan DW | 2015 | Hierarchical deconstruction of mouse olfactory sensory neurons: from whole mucosa to single-cell RNA-seq | https://www.ebi.ac.uk/ena/data/view/ERS715984 | European Nucleotide Archive, ERS715984 |

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
