## [Decision Letter]

Thank you for submitting your article "Tuning of Delta-Protocadherin Adhesion Through Combinatorial Diversity" for consideration by *eLife*. Your article has been reviewed by three peer reviewers, and the evaluation has been overseen by a Reviewing Editor and Didier Stainier as the Senior Editor. The reviewers have opted to remain anonymous.

The reviewers have discussed the reviews with one another and the Reviewing Editor has drafted this decision to help you prepare a revised submission.

This is an interesting and important paper describing an investigation of the function of δ-Pcdh proteins. They show that individual olfactory sensory neuron express as many as 7 δ-Pcdhs and address the principles that mediate cell adhesion. The authors point out interesting differences between δ-Pcdhs and previously described clustered Pcdhs, and they show that δ-Pcdhs can modify cell adhesion mediated by the clustered Pcdhs. All three reviewers believe that data presented significantly advance our understanding of Pcdhs, and provide interesting insights in the combinatorial interactions at the cell surface. However, all three reviews agree that the evidence Pcdh δ *cis* interaction is inadequate and the claim should be removed from the paper. I recommend publication if this claim is removed.

*Reviewer #1:*

This paper contains some interesting and publishable results concerning δ protocadherin expression patterns. However, a significant section of the paper is devoted to showing that these proteins undergo *cis* interactions but here, the strongest evidence seems to come from an analogy to clustered protocahderins. There is some indirect experimental support but the data presented can be interpreted in different ways. In the absence of real supporting evidence this section should be removed. At that point although there are questions as to the interest rises to the level of an *eLife* paper, I think the paper could in principle be accepted.

*Reviewer #2:*

This paper, by Lin and coworkers, shows that δ-Pcdhs are expressed in combinations in olfactory neurons and uses K562 cell aggregation assays to study δ-Pcdh cell-adhesive specificity and the effects on specificity of combinatorial expression. While these experiments are generally informative, the authors conclude from them that δ-Pcdhs interact promiscuously in *cis* to affect *trans* adhesion. While this is certainly possible, conclusions on *cis* interaction are, in my view, unsupported by the current data. Overall, this paper presents data of significant interest, but some of the conclusions need to be walked back until further experimental evidence is available.

The δ-Pcdhs represent a small family of cell-cell adhesion proteins. Importantly, these are divided into two subfamilies, δ-1 (containing 7 EC domains) and δ-2 (containing 6). While the authors do note this distinction in the Introduction, they drop it there and afterward refer only to δ-Pcdhs. I feel that these distinctions (i.e. which molecule is δ-1 and which is δ-2) should be continued throughout the manuscript.

The co-expression studies in OSNs are performed with two distinct methods, yielding similar results for each. The results are striking, showing co-expression of up to 7 δ-Pcdhs. When this analysis is performed for OSNs expressing particular odorant receptors, specific Pcdhs are associated with each OSN, with a small level of variability. Notably, it would be interesting to determine whether there is a preference for co-expression of δ-1 with δ-1 or δ-2 with δ-2 Pcdhs, but I do not think this analysis was performed.

The remainder of the paper is focused on results from K562 cell aggregation experiments. Aggregation behavior of single-transfectants shows that δ-Pcdhs have predominantly or exclusively homophilic recognition, does not require the cytoplasmic region, and is dependent on calcium. Further, using a centrifugation assay they reveal differences in the apparent affinities of various δ-Pcdhs. These results are straight-forward and valuable.

The remainder of the Results section is focused on the outcomes of mismatch coaggregation studies, with the goal of understanding the effects of combinatorial expression. These studies are interesting; while they test for coaggregation in a heterologous system, and it will thus difficult be certain how these will impact adhesion in vivo, I think they are informative for combinatorial behavior.

The authors explain the effects on aggregation of combinatorial expression by invoking the formation of cis dimers between co-expressed Pcdhs, analogous to observations for clustered protocadherins. To support this idea, they show that differentially tagged co-expressed δ-Pcdhs can be co-precipitated in pull-down experiments. Here is where I begin to have serious criticisms.

Overexpressed proteins often co-precipitate; such proteins could partake in shared junction or membrane structures, but have no direct interaction. The authors used a cytoplasmic GFP-FLAG as a negative control, but I would suggest that this is a poor control. Regardless, these data are too weak to infer direct *cis* association.

Further, it is not at all clear to me that *cis* associations need to be invoked to explain the results of the co-expression aggregation assays. While it is observed that co-expressing truncated Pcdhs has an impact on the fine structure of aggregates (e.g., interfacing rather than segregating), it is not clear to me why this would require *cis* interactions to explain. Overall, while this is an interesting paper, it would be significantly improved by reconsidering this important point.

*Reviewer #3:*

This study provides a very good analysis of the cell-cell binding specificities of the δ-protocadherin family. The authors begin by documenting overlapping and complex expression patterns of the δ-protocadherins in neural tissue, and hypothesize that they mediate combinatorial binding specificities. Although they do not examine whether these specifies have functional significance in vivo, they do demonstrate a complex pattern of combinatorial binding specificities using a quantitative operational assay involving cell surface expression and aggregation of normally non adherent cells. The experiments are carefully done, and include analysis of the effects of cell surface levels, and the use of a large number of protocadherins makes the findings robust.

I have two related concerns; one experimental and the other conceptual as a point of discussion. First, all the experiments were done with protocadherins lacking their cytoplasmic domains. Bonafide cell adhesion molecules typically require cytoplasmic domains to interact with the cytoskeleton and cell signaling molecules often require cytoplasmic interactions. Therefore it is important for the authors to test and confirm at least a few key examples of the specificity when the full length protocadherins are expressed. A conceptually related issue requires more discussion. The authors assume that because these molecules have cadherin repeats that they act as bonafide adhesion molecules that mediate the main physical interactions between cells and determine tissue architecture, like the classical cadherins. However, they may actually function as cell surface signaling molecules that mediate cell recognition events. There have been some in vivo studies suggesting that some δ protocadherins signal with minimal direct functions in physical cell adhesion; also juxtacrine cell signaling molecules can cause cell aggregation when expressed in non-adhesive cell lines. A more careful consideration of the actual potential in vivo roles of these molecules would make a better Discussion (and even better Introduction).

---

## [Author Response]

Reviewer #1:This paper contains some interesting and publishable results concerning δ-protocadherin expression patterns. However, a significant section of the paper is devoted to showing that these proteins undergo cis interactions but here, the strongest evidence seems to come from an analogy to clustered protocahderins. There is some indirect experimental support but the data presented can be interpreted in different ways. In the absence of real supporting evidence this section should be removed. At that point although there are questions as to the interest rises to the level of an eLife paper, I think the paper could in principle be accepted.Reviewer #2:This paper, by Lin and coworkers, shows that δ-Pcdhs are expressed in combinations in olfactory neurons and uses K562 cell aggregation assays to study δ-Pcdh cell-adhesive specificity and the effects on specificity of combinatorial expression. […] Overall, while this is an interesting paper, it would be significantly improved by reconsidering this important point.Reviewer #3:This study provides a very good analysis of the cell-cell binding specificities of the δ-protocadherin family. […] A more careful consideration of the actual potential in vivo roles of these molecules would make a better Discussion (and even better Introduction).

The consensus agreement among reviewers was the need to remove any mention of *cis* interactions among δ-protocadherins. We have done so. While we still believe such interactions are important and can influence δ-protocadherin adhesion in *trans*, we acknowledge additional experiments are needed. We are also removing Chunyan Wu from the author list, as her contributions revolved around these experiments.

Reviewer 2 asks that we place increased emphasis on the use of “δ-1” and “δ-2” to distinguish among the subfamilies. We have included additional text to accommodate this request. This reviewer also asked if there was any preference for δ-1 or δ-2 expression in our single cell analysis. Unfortunately, with only 50 cells, we did not have sufficient power to distinguish if one class or the other is preferentially expressed. We have added additional text to note this point, as well as included additional discussion of δ-1 and δ-2 subfamilies throughout the text.

Reviewer 3 asks whether or not full-length protocadherins produce the same effects as the ECTM constructs used in these experiments. We chose to use ECTM constructs for several reasons. First, from a practical point of view, expressing full-length constructs was more difficult than just the ECTM domain. However, there were other, theoretical reasons as well. While we acknowledge the importance of intracellular interactions, our goal was to perform a reductionist approach aimed at studying the adhesive properties of the deltas. Further, K562 cells are non-neuronal, non-adherent cells, and it is not clear that including the intracellular domain would replicate conditions within neurons. In addition, most δ family members have multiple isoforms. While the EC domain of each isoform is the same, the intracellular domains are not. As a result, how one would choose any particular isoform over another is not clear. We also showed in Figure 1—figure supplement 1 that the ECTM form of Pcdh1 possessed the same adhesive properties as the full-length version in K562 assays. This at least is consistent with the ECTM constructs reflecting the behavior of full-length plasmids. And finally, papers studying δ-protocadherin adhesion in general have used only the EC domain and not full-length constructs.

Nevertheless, we understand and acknowledge the reviewer’s concern, and appreciate the need to use full-length constructs where possible. We have therefore performed additional aggregation assays where possible using a full-length form, and found the same outcomes as those using the ECTM construct. We have noted this in the text as data not shown.